# Gated Integration of Low-Rank Adaptation
# for Continual Learning of Language Models

## Abstract

Continual learning, which requires the model to learn multiple tasks sequentially, is crucial for language models (LMs). Recently, low-rank adaptation (LoRA), one of the most representative parameter-efficient fine-tuning (PEFT) methods, has gained increasing attention in continual learning of LMs. However, most existing continual learning methods based on LoRA typically expand a new LoRA branch to learn each new task and force the new and old LoRA branches to contribute equally to old tasks, potentially leading to forgetting. In this work, we propose a new method, called gated integration of low-rank adaptation (GainLoRA), for continual learning of LMs. GainLoRA expands a new LoRA branch for each new task and introduces gating modules to integrate the new and old LoRA branches. Furthermore, GainLoRA leverages the new gating module to minimize the contribution from the new LoRA branch to old tasks, effectively mitigating forgetting and improving the model's overall performance. Experimental results on continual learning benchmarks demonstrate that GainLoRA outperforms existing state-of-the-art methods.

## 1. Introduction

Continual learning, which requires the model to learn multiple tasks sequentially, is crucial for language models (LMs) (Shi et al., 2024). Specifically, with extensive pre-trained knowledge and further fine-tuning strategies, existing LMs have demonstrated strong performance for a wide range of tasks (Brown et al., 2020; Zhang et al., 2022; Touvron et al., 2023). However, when learning multiple tasks sequentially, LMs may lose knowledge acquired from old tasks, resulting in a significant degradation in performance on old tasks. This phenomenon, known as catastrophic forgetting (Parisi et al., 2019; Luo et al., 2023; Wang et al., 2023a; 2024), highlights the need for developing effective continual learning methods for LMs. Existing continual learning methods can be categorized into two main categories. The first category (Razdaibiedina et al., 2023) assumes that task identities are available during inference, while the second category (Liang & Li, 2024; Zhao et al., 2024) tackles a more difficult and practical setting where task identities are unavailable during inference.

Recently, low-rank adaptation (LoRA) (Hu et al., 2022), one of the most representative parameter-efficient fine-tuning (PEFT) methods, has gained increasing attention in the continual learning of LMs (Wang et al., 2023a; Bohao et al., 2024). Specifically, by reparameterizing pre-trained weights in a low-rank form, LoRA updates only a limited number of parameters to adapt LMs to a downstream task, making the fine-tuning process much more efficient than updating all parameters of LMs (Han et al., 2024). This efficiency also benefits continual learning, making LoRA increasingly popular in continual learning of LMs.

Most existing continual learning methods based on LoRA (Liang & Li, 2024; Zhao et al., 2024) typically expand a new LoRA branch for learning each new task while freezing all old LoRA branches. In this way, they avoid forgetting caused by directly updating the LoRA parameters of old tasks (Qiao et al., 2024). However, to handle the practical continual learning scenario where task identities are unavailable at inference time, existing methods (Wang et al., 2023a; Liang & Li, 2024; Smith et al., 2024) based on LoRA integrate new and old LoRA branches through a simple addition. Consequently, they force the new and old LoRA branches to contribute equally to old tasks, which means that the new LoRA branch may cause a relatively large change in the model's output on old tasks. This leads to forgetting and degrades the model's overall performance in continual learning.

In this work, we propose a new method, called gated integration of low-rank adaptation (GainLoRA), for continual learning of LMs. The contributions of GainLoRA are listed as follows:

[1]Anonymous Institution, Anonymous City, Anonymous Region, Anonymous Country. Correspondence to: Anonymous Author <anon.email@domain.com>.

Preliminary work. Under review by the International Conference on Machine Learning (ICML). Do not distribute.

- GainLoRA expands a new LoRA branch to learn each new task and introduces gating modules to integrate the new and old LoRA branches.

- GainLoRA leverages the new gating module to minimize the contribution from the new LoRA branch to old tasks, effectively mitigating forgetting and improving the model's overall performance.

- Experimental results on continual learning benchmarks show that GainLoRA outperforms existing state-of-the-art continual learning methods.

## 2. Related Work and Preliminaries

### 2.1. Related Work

**Parameter-Efficient Fine-Tuning**  Parameter-efficient fine-tuning (PEFT) methods tune a limited number of parameters to adapt a pre-trained model for downstream tasks, showing much more efficiency than tuning all the parameters of the pre-trained model, especially for LMs. For example, Adapter (Houlsby et al., 2019) modifies the model architecture by introducing trainable modules into Transformer layers and tunes these modules for downstream tasks. Prompt-tuning (Lester et al., 2021) and Prefix-tuning (Li & Liang, 2021) insert learnable tokens into the input and tune them for downstream tasks. Low-rank adaptation (LoRA) (Hu et al., 2022) reparameterizes the original model parameters with low-rank matrices and tunes these matrices for downstream tasks. Although PEFT methods tune significantly fewer parameters than full fine-tuning, they can achieve comparable performance to full fine-tuning across a wide range of computer vision (CV) and natural language processing (NLP) tasks (Fu et al., 2022; Hu et al., 2022; Mahabadi et al., 2021; Zaken et al., 2022).

**Continual Learning**  There are three main types of continual learning methods, categorized as regularization-based methods, memory-based methods, and expansion-based methods. Regularization-based methods (Kirkpatrick et al., 2017; Aljundi et al., 2018; Jung et al., 2020; Smith et al., 2024) incorporate a regularization term to mitigate catastrophic forgetting. Memory-based methods (Aljundi et al., 2019a;b; Sun et al., 2022; Liang & Li, 2023a; Zhao et al., 2024) utilize memory mechanisms to preserve knowledge from old tasks. Expansion-based methods (Rusu et al., 2016; Hung et al., 2019; Li et al., 2019; Liang & Li, 2023b) mitigate catastrophic forgetting by introducing new parameters for learning new tasks while typically freezing old parameters.

Many continual learning methods (Aljundi et al., 2018; Arani et al., 2022; Liang & Li, 2023b) are designed to train models from scratch. Recent studies (Wang et al., 2022b; Smith et al., 2023b; Wang et al., 2023a; Liang & Li, 2024)

have shown that leveraging pre-trained models and PEFT strategies enables continual learning methods to achieve superior performance across tasks in both CV and NLP. For example, some methods (Wang et al., 2022b; Qin & Joty, 2022; Razdaibiedina et al., 2023) utilize prompt-tuning for continual learning. They either maintain independent prompts for each task or maintain a pool of prompts and select relevant ones from the pool for learning new tasks. Other methods (Wang et al., 2023a; Smith et al., 2023a; Liang & Li, 2024; Zhao et al., 2024) adopt LoRA for continual learning. Most of these methods expand a new LoRA branch to handle each new task while freezing old LoRA branches to mitigate catastrophic forgetting. However, they force the new and old LoRA branches to contribute equally to old tasks, potentially leading to forgetting.

### 2.2. Preliminaries

**Problem Definition**  We follow existing continual learning works (Wang et al., 2023a; Zhao et al., 2024) to formalize the problem definition for continual learning of LMs. Specifically, in continual learning, a sequence of tasks $\{\mathcal{T}_1, \mathcal{T}_2, ..., \mathcal{T}_T\}$ is presented to the model sequentially, where $T$ denotes the total number of tasks. The $t$-th task $\mathcal{T}_t$ consists of a training dataset $\mathcal{D}_t$. For any given sample $(\boldsymbol{x}_t, \boldsymbol{y}_t) \in \mathcal{D}_t$, $\boldsymbol{x}_t$ denotes an input sentence and $\boldsymbol{y}_t$ denotes the corresponding output. When learning the $t$-th new task, the model is required to mitigate catastrophic forgetting of the $t-1$ previously learned tasks.

Similar to existing continual learning works for LMs (Bohao et al., 2024; Zhao et al., 2024), we consider a more challenging continual learning setting defined by three key challenges: (1) the model is presented with a sequence of tasks spanning various types, such as dialogue generation, information extraction and so on; (2) the model is not provided with task identities at inference time; (3) the model must learn without access to real or synthetic samples from previously learned tasks.

**Low-Rank Adaptation**  LoRA (Hu et al., 2022) is a widely adopted PEFT method used for fine-tuning various pre-trained models, particularly LMs. Specifically, let $\boldsymbol{W} \in \mathbb{R}^{d_{out} \times d_{in}}$ represent a pre-trained weight in LMs, where $d_{in}$ and $d_{out}$ are the input and output dimensions, respectively. Instead of updating $\boldsymbol{W}$ directly, LoRA introduces an additional branch consisting of two matrices, $\boldsymbol{A} \in \mathbb{R}^{d_{out} \times r}$ and $\boldsymbol{B} \in \mathbb{R}^{r \times d_{in}}$, where $r \ll \min(d_{in}, d_{out})$. LoRA then modifies the forward propagation of this layer as $\boldsymbol{e} = (\boldsymbol{W} + \boldsymbol{A}\boldsymbol{B})\boldsymbol{h}$. Here, $\boldsymbol{h}$ and $\boldsymbol{e}$ denote the input and output, respectively. To ensure no initial impact on the pre-trained weights, $\boldsymbol{A}$ is initialized to $\boldsymbol{0}$, and $\boldsymbol{B}$ is initialized using a Gaussian distribution. During fine-tuning for downstream tasks, the pre-trained weight $\boldsymbol{W}$ remains frozen, and only the parameters $\boldsymbol{A}$ and $\boldsymbol{B}$ are fine-tuned.

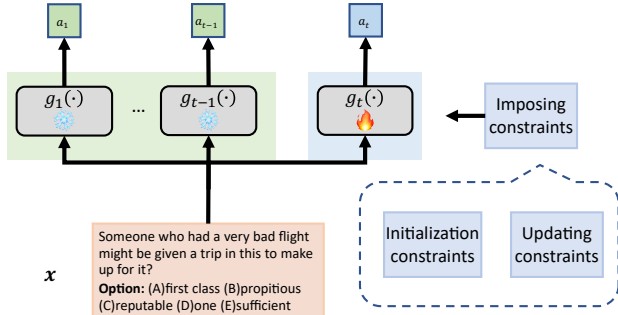

*Figure 1.* The expandable LoRA architecture of our GainLoRA for learning the $t$-th new task.

## 3. Methodology

Our GainLoRA employs an expandable LoRA architecture, which is illustrated in Figure 1. Specifically, before learning the $t$-th task ($1 \leq t \leq T$), GainLoRA first expands the LoRA architecture by introducing the $t$-th new branch with matrices $\boldsymbol{A}_t \in \mathbb{R}^{d_{out} \times r}$ and $\boldsymbol{B}_t \in \mathbb{R}^{r \times d_{in}}$. The new and old LoRA branches are then integrated as

$$\boldsymbol{W}_t = \boldsymbol{W}_{t-1} + a_t \boldsymbol{A}_t \boldsymbol{B}_t = \sum_{i=1}^{t} a_i \boldsymbol{A}_i \boldsymbol{B}_i, \quad (1)$$

where $a_i$ is an integration coefficient that determines the contribution of the $i$-th LoRA branch to the input $\boldsymbol{h}$. Note that $\boldsymbol{W}_{t-1}$ is a zero matrix when $t = 1$. As a result, the forward propagation in this layer is modified as

$$\boldsymbol{e} = (\boldsymbol{W} + \boldsymbol{W}_t)\boldsymbol{h}. \quad (2)$$

Finally, only the new LoRA branch (i.e. the $t$-th LoRA branch) is updated for the $t$-th new task, while all the old LoRA branches are frozen. After learning the $t$-th task, (2) is also used for inference across all test samples, thereby ensuring compatibility with the scenario where task identities are unavailable during inference.

Many existing continual learning methods based on LoRA (Wang et al., 2023a; Smith et al., 2024; 2023a; Liang & Li, 2024; Zhao et al., 2024) share a similar architecture to our method, as illustrated in Figure 1. However, these methods fix all coefficients $\{a_i\}_{i=1}^{t}$ in (1) to 1, forcing the new and old LoRA branches to contribute equally to old tasks. As a result, the new LoRA branch introduces a change of $\boldsymbol{A}_t \boldsymbol{B}_t \boldsymbol{h}$ to the output for inputs $\boldsymbol{h}$ associated with old tasks, potentially leading to forgetting (Qiao et al., 2024). Although some methods attempt to mitigate this forgetting

*Figure 2.* For each task $\mathcal{T}_i$, GainLoRA uses an independent gating module $g_i(\cdot)$ to generate integration coefficient $a_i$. Furthermore, during the learning of the $t$-th task, GainLoRA imposes constraints on the new gating module $g_t(\cdot)$.

by imposing regularization (Smith et al., 2024) or orthogonality constraints (Liang & Li, 2024) on the new LoRA branch, the fixed integration coefficients $\{a_i\}_{i=1}^{t}$ still limit their performance, as demonstrated by the experimental results presented in Section 4. Some method (Zhao et al., 2024) does not force the new and old LoRA branches to contribute equally to old tasks but relies on replaying synthetic old samples to mitigate forgetting, making it unsuitable for the scenario considered in this work.

Different from existing methods, GainLoRA introduces an independent gating module $g_i(\cdot)$ for each task $\mathcal{T}_i$ to generate the integration coefficients ($1 \leq i \leq T$). To mitigate the forgetting caused by the new task, GainLoRA leverages the gating module to minimize the contribution from the new LoRA branch to the old tasks. The details will be introduced in the following subsections.

### 3.1. Architecture of Gating Modules

As illustrated in Figure 2, given an input sample $\boldsymbol{x}$, the gating module $g_i(\cdot)$ generates the integration coefficient for the $i$-th LoRA branch, denoted as $a_i = g_i(\boldsymbol{x})$. The computation of $g_i(\cdot)$ is defined as

$$\begin{aligned} g_i(\boldsymbol{x}) &= f(\boldsymbol{W}_{i,L+1}\boldsymbol{p}_L), \\ \boldsymbol{p}_l &= \sigma(\boldsymbol{W}_{i,l}\boldsymbol{p}_{l-1}), \ l \in \{1, 2, ..., L\}, \\ \boldsymbol{p}_0 &= \text{Pool}(\text{Token}(\boldsymbol{x})). \end{aligned} \quad (3)$$

Here, $\text{Token}(\cdot)$ represents the tokenizer used in LMs to extract token embeddings from the input $\boldsymbol{x}$. $\text{Pool}(\cdot)$ denotes an average pooling operation applied to the token embeddings to produce a fixed-size vector. $\sigma(\cdot)$ denotes the non-linear activation function. $\boldsymbol{W}_{i,l}$ denotes the weight matrix for the $l$-th layer of $g_i(\cdot)$ ($1 \leq l \leq L+1$). In the final layer, $\boldsymbol{W}_{i,L+1}$ is a vector that maps the input vector $\boldsymbol{p}_{L+1}$ to a scalar. Following existing works with gating mecha-

nisms (Hochreiter & Schmidhuber, 1997; Cho, 2014), the function $f(\cdot)$ is designed to map a scalar to a value within $[0,1]$, that is, $f(\cdot) : \mathbb{R} \to [0,1]$.

Note that the input to gating modules is the same as that of LMs, denoted as $\boldsymbol{x}$, which differs from the input to LoRA in a specific layer, denoted as $\boldsymbol{h}$. During the learning of the $t$-th new task, only the new gating module $g_t(\cdot)$ is updated, while all the old gating modules $\{g_i(\cdot)\}_{i=1}^{t-1}$ remain frozen.

### 3.2. Minimizing the Contribution from the new LoRA branch to Old Tasks

GainLoRA minimizes the contribution from the new LoRA branch to old tasks by making $a_t = g_t(\boldsymbol{x})$ as close to 0 as possible for any input $\boldsymbol{x}$ from old tasks $\{\mathcal{T}_i\}_{i=1}^{t-1}$. However, since we focus on the scenario where no real or synthetic samples from old tasks are accessible, directly optimizing $g_t(\boldsymbol{x})$ to 0 is impractical. To overcome this challenge, GainLoRA imposes constraints on the new gating module $g_t(\cdot)$, implicitly guiding $g_t(\boldsymbol{x})$ to close to 0 and reduce the contribution of the new LoRA branch to old tasks.

In the following two subsections, we first describe the constraints imposed on the new gating module $g_t(\cdot)$ and explain how these constraints guide $g_t(\boldsymbol{x})$ close to 0 for any $\boldsymbol{x}$ from the old tasks. Then, we detail the implementation of these constraints during training.

#### 3.2.1. CONSTRAINTS ON NEW GATING MODULE

To formalize the constraints imposed on the new gating module $g_t(\cdot)$, we define the subspace spanned by the inputs to $\boldsymbol{W}_{t,l}$ ($1 \le l \le L+1$) from the previous $t-1$ tasks as:

$$\mathcal{M}_{t,l} = \mathrm{span}\{\boldsymbol{p}_{l-1} | \; \boldsymbol{p}_{l-1} \text{ is defined in (3)},$$
$$(\boldsymbol{x}, \boldsymbol{y}) \in \cup_{i=1}^{t-1} \mathcal{D}_i\}. \quad (4)$$

Note that subspaces $\{\mathcal{M}_{t,l}\}_{l=1}^{L+1}$ cannot be obtained directly due to the unavailability of samples from old tasks. However, by introducing additional constraints, $\{\mathcal{M}_{t,l}\}_{l=1}^{L+1}$ can be solved iteratively, which will be discussed in Section 3.2.2.

**Initialization Constraints**   Before learning the $t$-th task, the following constraints are imposed on the initialization of the new gating module $g_t(\cdot)$:

$$\mathrm{Init}(\boldsymbol{W}_{t,L+1}) \perp \mathcal{M}_{t,L+1}, \; f(0) = 0, \quad (5)$$

where $\mathrm{Init}(\boldsymbol{W}_{t,L+1})$ denotes the initialization of $\boldsymbol{W}_{t,L+1}$. These constraints ensure that for any sample $\boldsymbol{x}$ from the old tasks, the integration coefficient satisfies $a_t = g_t(\boldsymbol{x}) = f(\mathrm{Init}(\boldsymbol{W}_{t,L+1})\boldsymbol{p}_L) = 0$, where $\boldsymbol{p}_L$ is defined in (3). The second equality holds since $\boldsymbol{W}_{t,L+1} = \mathrm{Init}(\boldsymbol{W}_{t,L+1})$ before learning the $t$-th new task. The third equality holds because $f(0) = 0$ and $\boldsymbol{p}_L \in \mathcal{M}_{t,L+1}$ for any $\boldsymbol{x}$ from previous $t-1$ tasks.

**Updating Constraints**   During the learning of the $t$-th task, the following constraints are imposed on the updates to the new gating module $g_t(\cdot)$:

$$\Delta \boldsymbol{W}_{t,l} \perp \mathcal{M}_{t,l} \quad \text{for} \quad 1 \le l \le L+1, \quad (6)$$

where $\Delta \boldsymbol{W}_{t,l}$ denotes the update to $\boldsymbol{W}_{t,l}$. Based on existing studies (Wang et al., 2021; Liang & Li, 2023b; Qiao et al., 2024), the constraints in (6) ensure that $g_t(\boldsymbol{x})$ remains unchanged for inputs $\boldsymbol{x}$ from the old tasks during the learning of the $t$-th task. Formally, the following proposition holds:

**Proposition 3.1.** *If the constraints in (6) are satisfied, subspaces $\{\mathcal{M}_{t,l}\}_{l=1}^{L+1}$ remain unchanged during the learning of the $t$-th task. Furthermore, for any input $\boldsymbol{x}$ from the previous $t-1$ tasks, $g_t(\boldsymbol{x})$ remains unchanged during the learning of the $t$-th task.*

The proof of this proposition is provided in Appendix A.3. Since the initialization constraints in (5) ensure $g_t(\boldsymbol{x}) = 0$ before learning the $t$-th new task, $g_t(\boldsymbol{x}) = 0$ is preserved throughout the learning process if the updating constraints in (6) are satisfied.

The fact that subspaces $\{\mathcal{M}_{t,l}\}_{l=1}^{L+1}$ remain unchanged, as stated in Proposition 3.1, is essential for implementing the orthogonal constraints in (6). Specifically, as will be detailed in Section 3.2.2, orthonormal bases for the subspaces $\{\mathcal{M}_{t,l}\}_{l=1}^{L+1}$ are learned to enforce the orthogonal constraints in (5) and (6). Since the subspaces $\{\mathcal{M}_{t,l}\}_{l=1}^{L+1}$ remain unchanged during the learning of the $t$-th task, their orthonormal bases also remain unchanged, allowing them to be pre-computed before learning the $t$-th task, thus facilitating the implementation of orthogonal constraints in (5) and (6) throughout the learning process.

#### 3.2.2. IMPLEMENTATION OF CONSTRAINTS

There exist many functions $f(\cdot) : \mathbb{R} \to [0,1]$ satisfying $f(0) = 0$. In this work, we define $f(\cdot)$ as

$$f(b) = |2 \cdot \mathrm{sigmoid}(b) - 1|, \quad (7)$$

where $\mathrm{sigmoid}(\cdot)$ denotes the sigmoid function. Other functions $f(\cdot) : \mathbb{R} \to [0,1]$ that satisfy $f(0) = 0$ are also applicable, and experiments with different choices of $f(\cdot)$ are provided in Appendix C.3.1. Better model performance can be expected by designing more effective $f(\cdot)$, but this is not the focus of this paper.

Implementing the orthogonal constraints in (5) and (6) is challenging due to the lack of samples from previous $t-1$ tasks to approximate the subspaces $\{\mathcal{M}_{t,l}\}_{l=1}^{L+1}$. To address this issue, we further impose the following constraints on the initialization of $\boldsymbol{W}_{t,l}$ ($1 \le l \le L$):

$$\mathrm{Init}(\boldsymbol{W}_{t,l}) \leftarrow \boldsymbol{W}_{t-1,l}. \quad (8)$$

This strategy initializes the first $L$ layers of $g_t(\cdot)$ using the corresponding layers from the previous gating module $g_{t-1}(\cdot)$. As a result, the first $L$ layers of $g_t(\cdot)$ can be viewed as being initialized and starting their training at the beginning of the first task, continuing until the $t$-th task. Simultaneously, the first $L$ layers in $g_i(\cdot)$ serve as checkpoints, preserving the state of $g_t(\cdot)$ after learning the $i$-th task ($1 \leq i \leq t$). At this time, we can use existing method gradient projection memory (GPM) (Saha et al., 2021) to iteratively learn a set of matrices $\{M_{t,l}\}_{l=1}^{L+1}$, where the columns of $M_{t,l}$ contribute to a set of orthonormal bases of subspace $\mathcal{M}_{t,l}$. Details of GPM are provided in Appendix A.1. Then, before learning the $t$-th task, the following operation can be performed on $\text{Init}(W_{t,L+1})$:

$$\text{Init}(W_{t,L+1}) \leftarrow \text{Init}(W_{t,L+1}) \\ - M_{t,L+1}M_{t,L+1}^T \text{Init}(W_{t,L+1}). \quad (9)$$

According to existing works (Wang et al., 2021; Saha et al., 2021; Liang & Li, 2023b), $\text{Init}(W_{t,L+1})$ satisfies the constraints in (5) after the operation in (9). Similarly, during the learning of the $t$-th task, the following operation can be performed on $\{\Delta W_{t,l}\}_{l=1}^{L+1}$:

$$\Delta W_{t,l} \leftarrow \Delta W_{t,l} - M_{t,l}M_{t,l}^T \Delta W_{t,l}. \quad (10)$$

After this, $\{\Delta W_{t,l}\}_{l=1}^{L+1}$ satisfy the constraints in (6).

### 3.3. Updating the New LoRA Branch

Our GainLoRA aims to effectively integrate new and old LoRA branches while mitigating forgetting caused by the new LoRA branch on old tasks. Since GainLoRA does not impose specific update strategies for the new LoRA branch, it is inherently compatible with various existing continual learning methods that adopt similar LoRA architecture as our method and can update the new LoRA branch (Wang et al., 2023a; Liang & Li, 2024; Smith et al., 2024). Since these existing methods fix all integration coefficients $\{a_i\}_{i=1}^t$ to 1, combining our method with these existing methods can enhance their performance, as demonstrated in Section 4.

### 3.4. Whole Process of GainLoRA

Algorithm 1 outlines the whole process of our GainLoRA. Before learning the $t$-th new task $\mathcal{T}_t$, GainLoRA first expands the LoRA architecture by introducing the $t$-th new branch with matrices $A_t$ and $B_t$. Simultaneously, a new gating module $g_t(\cdot)$ is initialized through the operations specified in (7), (9) and (8) to ensure that the initialization constraints in (5) are satisfied. The new and old LoRA branches are then integrated using (1), and the forward propagation is modified as (2).

During the learning of the $t$-th task $\mathcal{T}_t$ with the corresponding dataset $\mathcal{D}_t$, our method follows existing methods (Wang

---

**Algorithm 1** GainLoRA for Continual Learning

**Input:** The data of different tasks $\{\mathcal{D}_t\}_{t=1}^T$.
**Output:** Learned LoRA parameters $\{(A_i, B_i)\}_{i=1}^T$ and gating modules $\{g_i(\cdot)\}_{i=1}^T$.
**for** $t$ in $1:T$ **do**
    Expand the $t$-th new LoRA branch with $A_t$ and $B_t$;
    Impose initialization constraints on the new gating module $g_t(\cdot)$ by (7), (8) and (9);
    Integrate new and old LoRA branches by (1);
    **for** $\mathcal{B}_t \subseteq \mathcal{D}_t$ **do**
        Compute the loss in (11) and the update of the parameters in the new LoRA branch and the new gating module;
        Impose updating constraints on the update of the new gating module by (6);
    **end for**
**end for**

---

et al., 2023a; Zhao et al., 2024) and computes the loss for the new task through

$$\mathcal{L}_t = \frac{1}{|\mathcal{D}_t|} \sum_{(\boldsymbol{x}_t, \boldsymbol{y}_t) \in \mathcal{D}_t} \sum_{j=1}^{|\boldsymbol{y}_t|} \log\left[P(y_{t,j}|\boldsymbol{x}_t, y_{t,1}, ..., y_{t,j-1})\right],$$

$$\quad (11)$$

where $\boldsymbol{y}_t = [y_{t,1}, y_{t,2}, ..., y_{t,|\boldsymbol{y}_t|}]$. Each time, GainLoRA samples a mini-batch $\mathcal{B}_t$ to minimize the loss in (11) by updating the new LoRA branch and the new gating module $g_t(\cdot)$. During this process, the projections defined in (10) are applied to the parameters of $g_t(\cdot)$, ensuring that the update constraints in (6) are satisfied.

Our GainLoRA introduces a new gating module for each new task, which inevitably incurs additional parameters and computational overhead when combined with other methods. Section 4 will demonstrate that the trainable parameters added by our method are limited, making the number of trainable parameters in our method comparable to other methods. Additionally, Appendix C.1 will demonstrate that the computational cost introduced by GainLoRA is minimal compared to the original LMs.

## 4. Experiments

### 4.1. Experimental Settings

**Datasets** Following existing continual learning methods (Razdaibiedina et al., 2023; Wang et al., 2023a; Zhao et al., 2024), we evaluate different methods on SuperNI (Wang et al., 2022a) and Long Sequence (Razdaibiedina et al., 2023) benchmarks. SuperNI benchmark includes various types of NLP tasks, including dialogue generation, information extraction, question answering, summarization, and sentiment analysis. Following the protocols

*Table 1.* Results on different task sequences with T5-large model. Results of methods with $^*$ are copied from existing paper (Zhao et al., 2024).

| Method | Order 1 | | Order 2 | | Order 3 | | Order 4 | |
|---|---|---|---|---|---|---|---|---|
| | AP↑ | FT↓ | AP↑ | FT↓ | AP↑ | FT↓ | AP↑ | FT↓ |
| L2P$^*$ (Wang et al., 2022b) | 15.18 | 3.65 | 10.27 | 12.24 | 58.61 | 15.43 | 57.34 | 17.82 |
| LFPT5$^*$ (Qin & Joty, 2022) | 39.03 | 9.85 | 29.70 | 19.08 | 66.62 | 13.60 | 67.40 | 11.99 |
| EPI$^*$ (Wang et al., 2023b) | - | - | - | - | 75.19 | 0.60 | 75.10 | 2.23 |
| MIGU+FT (Du et al., 2024) | - | - | - | - | 71.30 | 11.39 | 69.05 | 14.06 |
| SeqLoRA | 7.30 | 47.60 | 7.03 | 47.97 | 49.46 | 27.60 | 33.81 | 45.53 |
| IncLoRA (Wang et al., 2023a) | 12.33 | 41.93 | 16.65 | 36.56 | 61.19 | 13.63 | 62.46 | 15.92 |
| C-LoRA (Smith et al., 2024) | 22.69 | 24.25 | 32.81 | 11.60 | 66.83 | 8.64 | 61.86 | 14.18 |
| O-LoRA (Wang et al., 2023a) | 26.37 | 19.15 | 32.83 | 11.99 | 70.98 | 3.69 | 71.21 | 4.03 |
| GainLoRA (O-LoRA) | **47.84** | **2.26** | **46.84** | 2.91 | 73.37 | 3.02 | 76.01 | 2.49 |
| InfLoRA (Liang & Li, 2024) | 39.78 | 7.64 | 39.57 | 8.93 | 75.15 | 4.19 | 75.79 | 3.47 |
| GainLoRA (InfLoRA) | 46.21 | 2.40 | 46.44 | **2.61** | **78.01** | **0.77** | **77.54** | **1.25** |

of existing method (Zhao et al., 2024), three tasks are selected from each type, resulting in 15 tasks. These tasks are arranged into two different task sequences with different orders, referred to as Order 1 and Order 2. Long Sequence benchmark consists of 15 diverse classification tasks, which are similarly arranged into two task sequences with different orders, referred to as Order 3 and Order 4. More details about the benchmarks and task sequences are provided in Appendix B.

**Evaluation Metric**   We use $A_{j,i}$ to denote the model's performance on the $i$-th task once the model learns the $j$-th task. Specifically, $A_{j,i}$ represents accuracy for classification tasks and Rouge-L (Lin, 2004) for other types of tasks. Following traditional continual learning works (Chaudhry et al., 2019; Deng et al., 2021), we employ average performance (AP) and forgetting (FT) to evaluate the model's performance. The formulas for these two metrics are defined as

$$AP = \frac{1}{T} \sum_{i=1}^{T} A_{T,i},$$

$$FT = \frac{1}{T-1} \sum_{i=1}^{T-1} (\max_{l \in \{1,2,...,T-1\}} A_{l,i} - A_{T,i}), \quad (12)$$

where $T$ denotes the total number of tasks in the task sequence. AP evaluates the model's final performance, and FT quantifies the forgetting.

**Baselines**   We compare our method with state-of-the-art continual learning methods, including L2P (Wang et al., 2022b), LFPT5 (Qin & Joty, 2022), EPI (Wang et al., 2023b), MIGU (Du et al., 2024), IncLoRA (Wang et al., 2023a), C-LoRA (Smith et al., 2024), O-LoRA (Wang et al., 2023a), and InfLoRA (Liang & Li, 2024). Additionally, we introduce a simple baseline called SeqLoRA, which does not expand new LoRA branches but sequentially updates

old LoRA parameters for new tasks and lacks mechanism to mitigate forgetting.

**Implementation Details**   Following existing continual learning works (Ouyang et al., 2022; Wei et al., 2022; Wang et al., 2023a), all methods are implemented with instruction tuning (Ouyang et al., 2022) and optimized using AdamW (Loshchilov & Hutter, 2019). To ensure fair comparisons, for all the methods based on LoRA, we follow existing continual learning methods (Hu et al., 2022; Wang et al., 2023a; Zhao et al., 2024) by incorporating the LoRA architecture into the query and value components of the multi-head attention mechanism in each Transformer block. We use T5 (Raffel et al., 2020) and Llama-2 (Touvron et al., 2023) as the base architectures, aligning with the existing continual learning methods for LMs (Wang et al., 2023a; Zhao et al., 2024). Each experiment is repeated three times with different seeds, and the average result is reported. More details, such as the learning rate, batch size, and architecture of the gating modules in GainLoRA, are provided in Appendix B.2 and Appendix B.3.

### 4.2. Experimental Results

**Compare with Existing Methods**   We first follow existing works (Zhao et al., 2024; Du et al., 2024) and evaluate different continual learning methods using T5-Large. Since our method does not impose specific update strategies for the new LoRA branch, we adopt the same update strategies as the two state-of-the-art methods, O-LoRA (Wang et al., 2023a) and InfLoRA (Liang & Li, 2024). Note that these two methods leverage LoRA architecture similar to our method but fix all integration coefficients $\{a_i\}_{i=1}^{T}$ to 1. Details of these two methods are provided in Appendix A.2. We use GainLoRA (O-LoRA) and GainLoRA (InfLoRA) to respectively denote our methods adopting O-LoRA and InfLoRA to update the new LoRA branch. GainLoRA is also

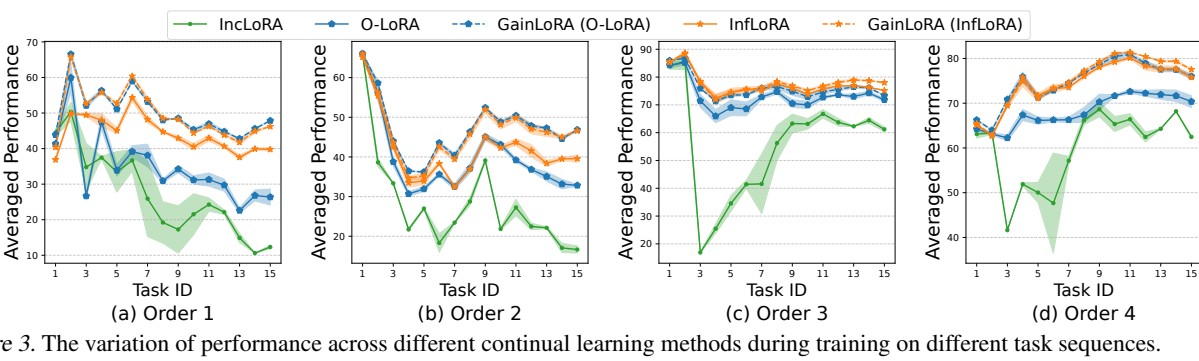

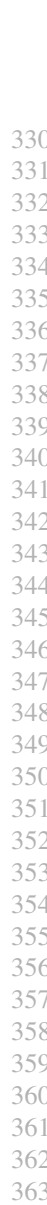

*Figure 3.* The variation of performance across different continual learning methods during training on different task sequences.

*Table 2.* The overall results on different task sequences with T5-XL model.

| Method | Order 1 | | Order 2 | | Order 3 | | Order 4 | |
|---|---|---|---|---|---|---|---|---|
| | AP↑ | FT↓ | AP↑ | FT↓ | AP↑ | FT↓ | AP↑ | FT↓ |
| O-LoRA (Wang et al., 2023a) | 36.50 | 11.42 | 40.64 | 6.37 | 73.77 | 2.70 | 76.19 | 3.56 |
| GainLoRA (O-LoRA) | **50.10** | 3.21 | 49.86 | 3.04 | 78.41 | 2.59 | 77.21 | 3.30 |
| InfLoRA (Liang & Li, 2024) | 45.61 | 5.60 | 45.85 | 5.10 | 80.22 | 2.09 | 79.43 | 1.71 |
| GainLoRA (InfLoRA) | 50.06 | **1.86** | **50.26** | **2.64** | **81.22** | **0.58** | **80.30** | **0.75** |

*Table 3.* The overall results on different task sequences with Llama-2-7B and Llama-2-13B.

| Method | Llama-2-7B | | | | Llama-2-13B | | | |
|---|---|---|---|---|---|---|---|---|
| | Order 1 | | Order 2 | | Order 1 | | Order 2 | |
| | AP↑ | FT↓ | AP↑ | FT↓ | AP↑ | FT↓ | AP↑ | FT↓ |
| O-LoRA (Wang et al., 2023a) | 39.37 | 15.84 | 37.55 | 20.23 | 43.92 | 14.15 | 40.05 | 19.53 |
| GainLoRA (O-LoRA) | 51.10 | 4.96 | **51.14** | 5.57 | 52.47 | 4.78 | 51.68 | 5.86 |
| InfLoRA (Liang & Li, 2024) | 42.93 | 11.23 | 39.94 | 15.00 | 43.64 | 14.85 | 45.74 | 10.61 |
| GainLoRA (InfLoRA) | **51.27** | **2.84** | 50.17 | **4.71** | **53.64** | **2.87** | **52.46** | **4.90** |

compatible with other methods that leverage expandable LoRA architecture shown in Figure 1, and we give some results in Appendix C.5.

The results are shown in Table 1. As we can see, our methods GainLoRA (O-LoRA) and GainLoRA (InfLoRA) outperform O-LoRA and InfLoRA in both AP and FT, respectively. This improvement demonstrates that fixing all coefficients $\{a_i\}_{i=1}^T$ to 1 leads to forgetting on old tasks, thereby limiting the performance of O-LoRA and InfLoRA. By effectively mitigating this forgetting, GainLoRA (O-LoRA) and GainLoRA (InfLoRA) achieve superior performance. Furthermore, our methods consistently achieve the best performance across all task sequences.

Figure 3 illustrates the variation in the average performance across all learned tasks for different methods throughout the continual learning process. As shown, GainLoRA consistently outperforms the performance of O-LoRA and InfLoRA throughout the whole training process.

**Scaling to Larger Model Architectures** To evaluate the effectiveness of our method on larger model architectures, we scale different LoRA-based continual learning meth-

ods to larger models, including T5-XL, Llama-2-7B, and Llama-2-13B. Table 2 and Table 3 present the results of different methods. As shown, across models of varying sizes, GainLoRA (O-LoRA) and GainLoRA (InfLoRA) consistently outperform O-LoRA and InfLoRA in terms of AP and FT, respectively. This demonstrates that GainLoRA effectively mitigates forgetting in the new LoRA branch across different model architectures.

**Trainable Parameters** We compare the number of trainable parameters across different methods for training on different task sequences. The results are shown in Figure 4, and the detailed computation of trainable parameters is provided in Appendix B.4.

As shown, GainLoRA (O-LoRA) and GainLoRA (InfLoRA) have more trainable parameters than O-LoRA and InfLoRA, respectively. This increase arises from the introduction of the trainable gating module in GainLoRA. However, the additional trainable parameters introduced by GainLoRA are much fewer than those in LoRA. Therefore, the total number of trainable parameters in GainLoRA (O-LoRA) and GainLoRA (InfLoRA) are comparable to that of O-LoRA and InfLoRA, respectively.

Table 4. Ablation study of GainLoRA with T5-Large and Llama-2-7B.

| Method | T5-Large | | | | Llama-2-7B | | | |
| --- | --- | --- | --- | --- | --- | --- | --- | --- |
| | Order 1 | | Order 2 | | Order 1 | | Order 2 | |
| | AP↑ | FT↓ | AP↑ | FT↓ | AP↑ | FT↓ | AP↑ | FT↓ |
| GainLoRA (O-LoRA) | **47.84** | **2.26** | **46.84** | **2.91** | **51.10** | **4.96** | **51.14** | **5.57** |
| No Initialization Constraints | 35.30 | 17.19 | 39.82 | 12.90 | 44.02 | 11.71 | 42.89 | 14.77 |
| No Updating Constraints | 23.01 | 30.32 | 24.96 | 28.14 | 33.74 | 23.06 | 34.71 | 22.36 |
| No Constraints | 26.32 | 26.00 | 30.63 | 22.37 | 34.48 | 23.46 | 36.87 | 21.24 |
| GainLoRA (InfLoRA) | **46.21** | **2.40** | **46.44** | **2.61** | **51.27** | **2.84** | **50.17** | **4.71** |
| No Initialization Constraints | 45.38 | 3.40 | 43.05 | 5.15 | 50.48 | 3.48 | 48.17 | 6.45 |
| No Updating Constraints | 37.69 | 10.94 | 38.85 | 9.31 | 48.52 | 5.68 | 47.85 | 7.00 |
| No Constraints | 36.75 | 12.18 | 41.00 | 6.66 | 49.10 | 6.07 | 45.77 | 8.70 |

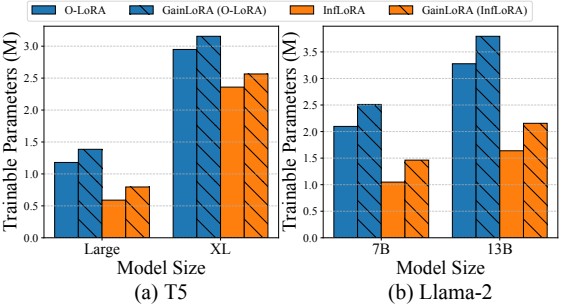

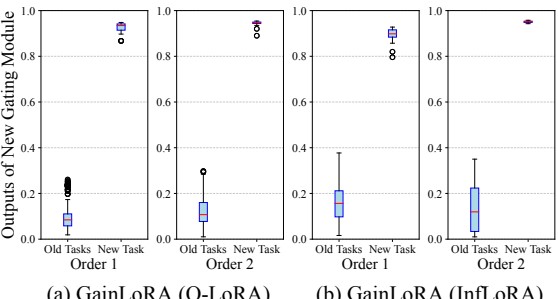

Figure 4. The number of trainable parameters in different continual learning methods with different model backbones on task sequences Order 1 and Order 2.

Figure 5. Outputs of new gating module in our GainLoRA on different task sequences with T5-Large.

**Ablation Study**    To verify the necessity of both the initialization and updating constraints introduced in Section 3.2.1, we define several variants of GainLoRA. The first variant, referred to as "No Initialization Constraints", removes the initialization constraints defined in (5). Specifically, it replaces $f(\cdot)$ defined in (7) with function $\mathrm{sigmoid}(\cdot)$ and eliminates the operation in (9) while keeping all other components unchanged. The second variant, referred to as "No Updating Constraints", removes the updating constraints defined in (6) by eliminating the operations in (10) while preserving all other components of GainLoRA. The third variant, referred to as "No Constraints", follows "No Initialization Constraints" and "No Updating Constraints" to remove both the initialization and updating constraints.

Table 4 presents the experimental results of these variants. As shown, none of these variants perform as well as our GainLoRA, indicating the critical role of both the initialization constraints and updating constraints in our GainLoRA.

**Distribution of Outputs in New Gating Module**    To demonstrate that our GainLoRA effectively minimizes the contribution from the new LoRA branches to old tasks, we analyze the output distributions of the new gating modules. Specifically, after training on the final task (i.e., the 15-th task) in the task sequences, the 15-th task corresponds to

the new task, and its associated gating module $g_{15}(\cdot)$ serves as the new gating module.

We obtain the outputs of the new gating module $g_{15}(\cdot)$ on the samples from old and new tasks, respectively. Then, we analyze their distributions in Figure 5. As shown, the outputs of $g_{15}(\cdot)$ for the samples from old tasks are concentrated around 0, effectively minimizing the contribution from the new LoRA branch to old tasks. Furthermore, GainLoRA does not constrain the outputs of $g_{15}(\cdot)$ for the samples from the new task. As a result, the outputs of $g_{15}(\cdot)$ for the samples from the new task are distributed near 1, enabling the model to effectively learn the new task.

## 5. Conclusion

In this work, we propose a new method, called GainLoRA, for continual learning of language models. GainLoRA expands a new LoRA branch for each new task and introduces gating modules to integrate the new and old LoRA branches. Furthermore, GainLoRA leverages the new gating module to minimize the contribution of the new LoRA branch to old tasks, effectively mitigating forgetting and improving the model's overall performance. Experimental results on continual learning benchmarks demonstrate that GainLoRA outperforms existing state-of-the-art methods.

## Impact Statement

This paper aims to contribute to the advancement of the machine learning field. While our work may have various societal implications, we do not find any that requires specific emphasis.

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

# A. More Details of Methods

## A.1. Gradient Projection Memory

We initialize the first $L$ layers of $g_t(\cdot)$ using the corresponding layers from the previous gating module $g_{t-1}(\cdot)$. Therefore, the first $L$ layers of $g_t(\cdot)$ can be viewed as being initialized at the beginning of the first task and continue their training until the $t$-th task. Additionally, the first $L$ layers in $g_i(\cdot)$ serve as checkpoints, preserving the state of $g_t(\cdot)$ after learning the $i$-th task ($1 \leq i \leq t$). At this time, existing method gradient projection memory (GPM) (Saha et al., 2021) can be used to learn matrices $\{M_{t,l}\}_{l=1}^{L+1}$, where the columns of $M_{t,l}$ approximate the orthonormal bases of the subspace $\mathcal{M}_{t,l}$. Specifically, when $t = 1$, since there is no old task, $\mathcal{M}_{1,l}$ is a null space and $M_{1,l}$ is a zero matrix. After learning the $t$-th new task, GPM expands $\mathcal{M}_{t,l}$ to $\mathcal{M}_{t+1,l}$ by first computing the input matrix $H_{t,l}$ where each column of $H_{t,l}$ represents an input to the $l$-th layer. Then, the component of $H_{t,l}$ already in $\mathcal{M}_{t,l}$ is removed by

$$\widehat{H}_{t,l} = H_{t,l} - M_{t,l}(M_{t,l})^T H_{t,l}. \tag{13}$$

Next, singular value decomposition (SVD) is performed on $\widehat{H}_{t,l}\widehat{H}_{t,l}^T$, which is decomposed as $\widehat{U}_{t,l}\widehat{\Sigma}_{t,l}\widehat{U}_{t,l}^T$. Then, $u$ new orthonormal bases $u_1, ..., u_u$ are chosen from the columns of $\widehat{U}_{t,l}$, where $u$ is the minimum number satisfying the following criteria for a given threshold $\epsilon_{th}$:

$$||(\widehat{H}_{t,l})_u||_F^2 + ||M_{t,l}(M_{t,l})^T H_{t,l}||_F^2 \geq \epsilon_{th}||H_{t,l}||_F^2. \tag{14}$$

Here, $(\widehat{H}_{t,l})_u$ denotes the components of $\widehat{H}_{t,l}$ corresponding to the top-$u$ singular values. Then, the orthonormal bases of subspace $\mathcal{M}_{t+1,l}$ are obtained by augmenting the orthonormal bases of subspace $\mathcal{M}_{t,l}$ with the new orthogonal vectors $u_1, ..., u_u$, resulting in $M_{t+1,l} = [M_{t,l}, u_1, ..., u_u]$.

## A.2. More Details of O-LoRA and InfLoRA

**O-LoRA**  O-LoRA (Wang et al., 2023a) ensures that the new LoRA branch remains orthogonal to all the old LoRA branches. Specifically, during the learning of the $t$-th new task with the $t$-th LoRA branch $(A_t, B_t)$, O-LoRA computes the inner product between the new and old LoRA branches as

$$O_{i,t} = B_i^T B_t \quad \text{for } 1 \leq i \leq t - 1 \tag{15}$$

Then, the loss function of O-LoRA is defined as

$$\frac{1}{|\mathcal{D}_t|} \sum_{(\boldsymbol{x}_t, \boldsymbol{y}_t) \in \mathcal{D}_t} \sum_{j=1}^{|\boldsymbol{y}_t|} \log\left[P(y_{t,j}|\boldsymbol{x}_t, y_{t,1}, ..., y_{t,j-1})\right] + \lambda \sum_{i=1}^{t-1} \sum_{j,k} ||O_{i,t}[j,k]||_2^2 \tag{16}$$

For further details on O-LoRA, we refer readers to the original paper (Wang et al., 2023a).

**InfLoRA**  InfLoRA (Liang & Li, 2024) ensures orthogonality between the new LoRA branch and the gradients of old tasks. Specifically, it shows that only fine-tuning the down-projection matrix $A_t$ in the new LoRA branch is equivalent to directly fine-tuning the pre-trained weights within a subspace spanned by the rows of $B_t$. Therefore, before learning the $t$-th task, InfLoRA designs $B_t$ to be orthogonal to the gradients of the old tasks. During the learning of the $t$-th task, InfLoRA only tunes $A_t$ in the new LoRA branch while freezing $B_t$ and all the old LoRA branches. For further details on InfLoRA, we refer readers to the original paper (Liang & Li, 2024).

## A.3. Proof of Proposition 3.1

**Proposition A.1.** *If the constraints in (6) are satisfied, subspaces $\{\mathcal{M}_{t,l}\}_{l=1}^{L+1}$ remain unchanged during the learning of the $t$-th task. Furthermore, for any input $\boldsymbol{x}$ from the previous $t - 1$ tasks, $g_t(\boldsymbol{x})$ remains unchanged during the learning of the $t$-th task.*

*Proof.* For any $\boldsymbol{x}$ from previous $t - 1$ tasks, we rewrite $\boldsymbol{g}_t(\boldsymbol{x})$ as

$$\begin{aligned} g_t(\boldsymbol{x}) &= f(\boldsymbol{W}_{t,L+1}\boldsymbol{p}_L), \\ \boldsymbol{p}_l &= \sigma(\boldsymbol{W}_{t,l}\boldsymbol{p}_{l-1}), \ l \in \{1, 2, ..., L\}, \\ \boldsymbol{p}_0 &= \text{Pool}(\text{Token}(\boldsymbol{x})). \end{aligned} \tag{17}$$

Since $\boldsymbol{p}_0 = \text{Pool}(\text{Token}(\boldsymbol{x}))$ is unrelated to the parameters of the new gating module $g_t(\cdot)$, $\boldsymbol{p}_0$ does not change with the update of $g_t(\cdot)$. Since $\mathcal{M}_{t,1}$ is spanned by $\boldsymbol{p}_0$, $\mathcal{M}_{t,1}$ remains unchanged during the learning of the $t$-th task.

Suppose that we have proven that $\boldsymbol{p}_{l-1}$ does not change with the update of the new gating module $g_t(\cdot)$ ($1 \leq l \leq L$). Since $\mathcal{M}_{t,l}$ is spanned by $\boldsymbol{p}_{l-1}$, $\mathcal{M}_{t,l}$ remains unchanged during the learning of the $t$-th task. At this point, $\boldsymbol{p}_l$ can be expressed as

$$\boldsymbol{p}_l = \sigma((\text{Init}(\boldsymbol{W}_{t,l}) + \Delta \boldsymbol{W}_{t,l})\boldsymbol{p}_{l-1}) = \sigma(\text{Init}(\boldsymbol{W}_{t,l})\boldsymbol{p}_{l-1}). \tag{18}$$

Here, the second equality holds since $\boldsymbol{p}_{l-1} \in \mathcal{M}_{t,l}$ and $\Delta \boldsymbol{W}_{t,l} \perp \mathcal{M}_{t,l}$. Therefore, $\boldsymbol{p}_l$ does not change with the update of the new gating module $g_t(\cdot)$ ($1 \leq l \leq L$). Since $\mathcal{M}_{t,l+1}$ is spanned by $\boldsymbol{p}_l$, $\mathcal{M}_{t,l+1}$ remains unchanged during the learning of the $t$-th task.

Furthermore, during the learning of the $t$-th task, $g_t(\boldsymbol{x})$ can be expressed as

$$\boldsymbol{g}_t(\boldsymbol{x}) = f((\text{Init}(\boldsymbol{W}_{t,L+1}) + \Delta \boldsymbol{W}_{t,L+1})\boldsymbol{p}_L) = f(\text{Init}(\boldsymbol{W}_{t,L+1})\boldsymbol{p}_L). \tag{19}$$

Here, the second equality holds since $\boldsymbol{p}_L \in \mathcal{M}_{t,L+1}$ and $\Delta \boldsymbol{W}_{t,L+1} \perp \mathcal{M}_{t,L+1}$. □

## B. More Details of Experimental Settings

### B.1. More Details of Datasets

Table 5 and Table 6 show the details of Long Sequence Benchmark and SuperNI Benchmark, respectively. Long Sequence Benchmark consists of 15 classification tasks while SuperNI Benchmark consists of various NLP tasks, including dialogue generation, information extraction, question answering, summarization, and sentiment analysis.

*Table 5.* Details of different tasks in Long Benchmark.

| Dataset name | Category | Domain | Task Type | Metric |
|---|---|---|---|---|
| Yelp | CL Benchmark | sentiment analysis | Yelp reviews | Accuracy |
| Amazon | CL Benchmark | sentiment analysis | Amazon reviews | Accuracy |
| DBpedia | CL Benchmark | topic classification | Wikipedia | Accuracy |
| Yahoo | CL Benchmark | topic classification | Yahoo Q&A | Accuracy |
| AG News | CL Benchmark | topic classification | news | Accuracy |
| MNLI | GLUE | natural language inference | various | Accuracy |
| QQP | GLUE | paraphrase detection | Quora | Accuracy |
| RTE | GLUE | natural language inference | news, Wikipedia | Accuracy |
| SST-2 | GLUE | sentiment analysis | movie reviews | Accuracy |
| WiC | SuperGLUE | word sense disambiguation | lexical databases | Accuracy |
| CB | SuperGLUE | natural language inference | various | Accuracy |
| COPA | SuperGLUE | question and answering | blogs, encyclopedia | Accuracy |
| BoolQA | SuperGLUE | boolean question and answering | Wikipedia | Accuracy |
| MultiRC | SuperGLUE | question and answering | various | Accuracy |
| IMDB | SuperGLUE | sentiment analysis | movie reviews | Accuracy |

The task sequences are constructed using Long Sequence Benchmark and SuperNI Benchmark. The details of different task sequences are presented in Table 7.

### B.2. More Implementation Details

Following existing continual learning works (Ouyang et al., 2022; Wang et al., 2023a; Wei et al., 2022), all methods are implemented using instruction tuning (Ouyang et al., 2022). Experiments are conducted on NVIDIA RTX A6000 GPUs with AdamW (Loshchilov & Hutter, 2019) as the optimizer. For T5-Large and T5-XL, their relatively smaller model sizes allow

*Table 6.* Details of different tasks in SuperNI Benchmark.

| Dataset name | Task Type | Metric |
|---|---|---|
| Task639_multi_woz_user_utterance_generation | summarization | Rouge-L |
| Task1590_diplomacy_text_generation | summarization | Rouge-L |
| Task1729_personachat_generate_next | summarization | Rouge-L |
| Task181_outcome_extraction | information extraction | Rouge-L |
| Task748_glucose_reverse_cause_event_detection | information extraction | Rouge-L |
| Task1510_evalution_relation_extraction | information extraction | Rouge-L |
| Task002_quoref_answer_generation | dialogue generation | Rouge-L |
| Task073_commonsenseqa_answer_generation | dialogue generation | Rouge-L |
| Task591_sciq_answer_generation | dialogue generation | Rouge-L |
| Task511_reddit_tifu_long_text_summarization | question answering | Rouge-L |
| Task1290_xsum_summarization | question answering | Rouge-L |
| Task1572_samsum_summary | question answering | Rouge-L |
| Task363_sst2_polarity_classification | sentiment analysis | Accuracy |
| Task875_emotion_classification | sentiment analysis | Accuracy |
| Task1687_sentiment140_classification | sentiment analysis | Accuracy |

*Table 7.* The order of different task sequences for experiments.

| Benchmark | Order | Task Sequence |
|---|---|---|
| SuperNI Benchmark | 1 | task1572 → task363 → task1290 → task181 → task002 → task1510 → task639 → task1729 → task073 → task1590 → task748 → task511 → task591 → task1687 → task875 |
| | 2 | task748 → task073 → task1590 → task639 → task1572 → task1687 → task591 → task363 → task1510 → task1729 → task181 → task511 → task002 → task1290 → task875 |
| CL Benchmark | 3 | MNLI → CB → WiC → COPA → QQP → BoolQA → RTE → IMDB → Yelp → Amazon → SST-2 → DBpedia → AG News → MultiRC → Yahoo |
| | 4 | Yelp → Amazon → MNLI → CB → COPA → QQP → RTE → IMDB → SST-2 → DBpedia → AG News → Yahoo → MultiRC → BoolQA → WiC |

experiments to be performed on a single A6000 GPU with gradient accumulation. For Llama-2-7B and Llama-2-13B, data parallelism with DeepSpeed ZeRO-2 (Rasley et al., 2020) is prioritized across multiple A6000 GPUs. FlashAttention-2 (Dao, 2024) is employed to reduce memory usage during training, ensuring sufficient GPU memory to enable DeepSpeed ZeRO-2 whenever possible. However, if the sequence lengths of certain tasks are too long to enable DeepSpeed ZeRO-2 even with FlashAttention-2, DeepSpeed ZeRO-3 is utilized to handle these tasks.

To ensure fair comparisons, for all the methods based on LoRA, we follow existing continual learning methods (Hu et al., 2022; Wang et al., 2023a; Zhao et al., 2024) by integrating the LoRA architecture into the query and value components of the multi-head attention mechanism in each Transformer block. Following existing works (Wang et al., 2023a; Zhao et al., 2024), for all the methods based on LoRA, the rank of a single LoRA branch is set to 4 for Order 1 and Order 2, and 8 for Order 3 and Order 4. We also vary the rank in LoRA branches and show the results in Appendix C.4.

For our methods, the global batch size is set to 32 across all model backbones. The learning rate is set to 3e-4 for T5 backbones and 5e-5 for Llama backbones. Each task is trained for 100 epochs with T5 backbones and 50 epochs with Llama backbones. For baselines, we follow their official implementations to set the hyperparameters, making the comparison as fair as possible. If this does not achieve the expected performance, we perform a hyperparameter search for the learning rate

and batch size.

### B.3. More Details about the Architecture of the Gating Module

The architecture of the gating module $g_i(\cdot)$ can be represented as

$$
\begin{aligned}
g_i(\boldsymbol{x}) &= f(\boldsymbol{W}_{i,L+1}\boldsymbol{p}_L), \\
\boldsymbol{p}_l &= \sigma(\boldsymbol{W}_{i,l}\boldsymbol{p}_{l-1}), \; l \in \{1, 2, ..., L\}, \\
\boldsymbol{p}_0 &= \text{Pool}(\text{Token}(\boldsymbol{x})).
\end{aligned}
\tag{20}
$$

Non-linear activation function $\sigma(\cdot)$ is set to SiLU (Elfwing et al., 2018). For all experiments, unless otherwise stated, $L$ is set to 2. In other words, the gating module $g_i(\cdot)$ has three layers. For T5-Large and T5-XL, the parameters in the $i$-th gating module $g_i(\cdot)$ are $\boldsymbol{W}_{i,1} \in \mathbb{R}^{100 \times d}$, $\boldsymbol{W}_{i,2} \in \mathbb{R}^{d \times 100}$ and $\boldsymbol{W}_{i,3} \in \mathbb{R}^{1 \times d}$. For Llama-2-7B and Llama-2-13B, the parameters in the $i$-th gating module $g_i(\cdot)$ are $\boldsymbol{W}_{i,1} \in \mathbb{R}^{50 \times d}$, $\boldsymbol{W}_{i,2} \in \mathbb{R}^{d \times 50}$ and $\boldsymbol{W}_{i,3} \in \mathbb{R}^{1 \times d}$. Here, $d$ denotes the dimension of the embeddings. For different models, $d$ is 1024 for T5-Large and T5-XL, 4096 for Llama-2-7B, and 5120 for Llama-2-13B.

Additionally, we investigate the influence of the architecture of gating module on the performance of our method. Results are provided in Appendix C.3.

### B.4. Computation of Trainable Parameters

To ensure fair comparisons, we set the same rank for each LoRA branch across all continual learning methods based on the expandable LoRA architectures shown in Figure 1. Additionally, for all the methods based on LoRA, the LoRA modules are incorporated into the query and value components of the multi-head attention mechanism within each Transformer block.

#### B.4.1. COMPUTATION OF TRAINABLE PARAMETERS IN T5-LARGE

In T5-Large, the projection weights for the query and value components have shapes $\boldsymbol{W}_q, \boldsymbol{W}_v \in \mathbb{R}^{1024 \times 1024}$. The model consists of 24 self-attention modules in the encoder, 24 self-attention modules in the decoder, and 24 cross-attention modules in the decoder, resulting in a total of $(24 + 24 + 24) * 2 = 144$ pre-trained weights that incorporate the LoRA architecture.

During the learning of the $t$-th new task, O-LoRA updates the parameters $\boldsymbol{A}_t \in \mathbb{R}^{1024 \times r}$ and $\boldsymbol{B}_t \in \mathbb{R}^{r \times 1024}$, resulting in $1024 * r * 144 + r * 1024 * 144 = 294912r$ trainable parameters. When $r = 4$, the number of trainable parameters in O-LoRA is $294912 * 4 = 1179648 = 1.18M$. InfLoRA only updates the parameters $\boldsymbol{A}_t \in \mathbb{R}^{1024 \times r}$, resulting in $1024 * r * 144 = 147456r$ trainable parameters. When $r = 4$, the number of trainable parameters in InfLoRA is $147456r = 589824 = 0.59M$.

GainLoRA introduces an additional new gating module $g_t(\cdot)$ with parameters $\boldsymbol{W}_{t,1} \in \mathbb{R}^{100 \times 1024}$, $\boldsymbol{W}_{t,2} \in \mathbb{R}^{1024 \times 100}$ and $\boldsymbol{W}_{t,3} \in \mathbb{R}^{1 \times 1024}$. Therefore, the number of trainable parameters in GainLoRA (O-LoRA) is $1179648 + 1024 * 100 + 1024 * 100 + 1024 = 1385472 = 1.39M$. The number of trainable parameters in GainLoRA (InfLoRA) is $589824 + 1024 * 100 + 1024 * 100 + 1024 = 795648 = 0.80M$.

#### B.4.2. COMPUTATION OF TRAINABLE PARAMETERS IN T5-XL

In T5-XL, the projection weights for the query and value components have shapes $\boldsymbol{W}_q, \boldsymbol{W}_v \in \mathbb{R}^{4096 \times 1024}$. The model architecture is similar to T5-Large, with 144 pre-trained weights incorporating LoRA.

During the learning of the $t$-th new task, O-LoRA updates the parameters $\boldsymbol{A}_t \in \mathbb{R}^{4096 \times r}$ and $\boldsymbol{B}_t \in \mathbb{R}^{r \times 1024}$, resulting in is $4096 * r * 144 + r * 1024 * 144 = 737280r$ trainable parameters. When $r = 4$, O-LoRA has $737280 * 4 = 2949120 = 2.95M$ trainable parameters. InfLoRA only updates $\boldsymbol{A}_t \in \mathbb{R}^{4096 \times r}$, resulting in $4096 * r * 144 = 589824r$ trainable parameters. When $r = 4$, InfLoRA has $589824 * 4 = 2359296 = 2.36M$ trainable parameters.

GainLoRA introduces the same new gating module $g_t(\cdot)$ as in T5-Large, with parameters $\boldsymbol{W}_{t,1} \in \mathbb{R}^{100 \times 1024}$, $\boldsymbol{W}_{t,2} \in \mathbb{R}^{1024 \times 100}$ and $\boldsymbol{W}_{t,3} \in \mathbb{R}^{1 \times 1024}$. Thus, the total number of trainable parameters for GainLoRA (O-LoRA) is $2949120 + 1024 * 100 + 1024 * 100 + 1024 = 3154944 = 3.15M$. The total number of trainable parameters in GainLoRA (InfLoRA) is $2359296 + 1024 * 100 + 1024 * 100 + 1024 = 2565120 = 2.57M$.

*Table 8.* FLOPs and MACs for different models.

| | Method | Input Shape (batch,length) | FLOPs (G) | MACs (G) |
|---|---|---|---|---|
| T5-Large | Original | (1,128) | 194.25 | 97.1 |
| | GainLoRA (O-LoRA) | (1,128) | 198.79 | 99.37 |
| | GainLoRA (InfLoRA) | (1,128) | 198.79 | 99.37 |
| T5-XL | Original | (1,128) | 751.7 | 375.78 |
| | GainLoRA (O-LoRA) | (1,128) | 763.03 | 381.45 |
| | GainLoRA (InfLoRA) | (1,128) | 763.03 | 381.45 |
| Llama-2-7B | Original | (1,128) | 1701.07 | 850.5 |
| | GainLoRA (O-LoRA) | (1,128) | 1709.14 | 854.53 |
| | GainLoRA (InfLoRA) | (1,128) | 1709.14 | 854.53 |
| Llama-2-13B | Original | (1,128) | 3291.66 | 1645.79 |
| | GainLoRA (O-LoRA) | (1,128) | 3304.26 | 1652.09 |
| | GainLoRA (InfLoRA) | (1,128) | 3304.26 | 1652.09 |

### B.4.3. COMPUTATION OF TRAINABLE PARAMETERS IN LLAMA-2-7B

In Llama-2-7B, the projection weights for the query and value components have shapes $W_q, W_v \in \mathbb{R}^{4096 \times 4096}$. The model contains 32 self-attention modules, resulting in $32 * 2 = 64$ pre-trained weights that incorporate the LoRA architecture.

During the learning of the $t$-th new task, O-LoRA updates the parameters $A_t \in \mathbb{R}^{4096 \times r}$ and $B_t \in \mathbb{R}^{r \times 4096}$, resulting in $4096 * r * 64 + r * 4096 * 64 = 524288r$ trainable parameters. When $r = 4$, the number of trainable parameters in O-LoRA is $524288 * 4 = 2097152 = 2.10M$. InfLoRA only updates the parameters $A_t \in \mathbb{R}^{4096 \times r}$, resulting in $4096 * r * 64 = 262144r$ trainable parameters. When $r = 4$, the number of trainable parameters in InfLoRA is $262144 * 4 = 1048576 = 1.05M$.

GainLoRA introduces a new gating module $g_t(\cdot)$ with parameters $W_{t,1} \in \mathbb{R}^{50 \times 4096}$, $W_{t,2} \in \mathbb{R}^{4096 \times 50}$ and $W_{t,3} \in \mathbb{R}^{1 \times 4096}$. Therefore, the number of trainable parameters in GainLoRA (O-LoRA) is $2097152 + 4096 * 50 + 4096 * 50 + 4096 = 2510848 = 2.51M$. The number of trainable parameters in GainLoRA (InfLoRA) is $1048576 + 4096 * 50 + 4096 * 50 + 4096 = 1462272 = 1.46M$.

### B.4.4. COMPUTATION OF TRAINABLE PARAMETERS IN LLAMA-2-13B

In Llama-2-13B, the projection weights for the query and value components have shapes $W_q, W_v \in \mathbb{R}^{5120 \times 5120}$. The model contains 40 self-attention modules, resulting in $40 * 2 = 80$ pre-trained weights that incorporate the LoRA architecture.

During the learning of the $t$-th new task, O-LoRA updates the parameters $A_t \in \mathbb{R}^{5120 \times r}$ and $B_t \in \mathbb{R}^{r \times 5120}$, resulting in $5120 * r * 80 + r * 5120 * 80 = 819200r$ trainable parameters. When $r = 4$, the number of trainable parameters in O-LoRA is $819200 * 4 = 3276800 = 3.28M$. InfLoRA only updates the parameters $A_t \in \mathbb{R}^{5120 \times r}$, resulting in $5120 * r * 80 = 409600r$ trainable parameters. When $r = 4$, the number of trainable parameters in InfLoRA is $409600 * 4 = 1638400 = 1.64M$.

GainLoRA introduces a new gating module $g_t(\cdot)$ with parameters $W_{t,1} \in \mathbb{R}^{50 \times 5120}$, $W_{t,2} \in \mathbb{R}^{5120 \times 50}$ and $W_{t,3} \in \mathbb{R}^{1 \times 5120}$. Therefore, the number of trainable parameters in GainLoRA (O-LoRA) is $3276800 + 5120 * 50 + 5120 * 50 + 5120 = 3793920 = 3.79M$. The number of trainable parameters in GainLoRA (InfLoRA) is $1638400 + 5120 * 50 + 5120 * 50 + 5120 = 2155520 = 2.16M$.

## C. More Experimental Results

### C.1. Discussing Computational Costs Introduced by GainLoRA

Existing methods, such as O-LoRA and InfLoRA, adopt the expandable LoRA architecture shown in Figure 1 and fix the integration coefficients $\{a_i\}_{i=1}^{T}$ to 1, allowing the model to merge the expanded LoRA branches into the pre-trained matrix at inference time, thereby avoiding additional computational costs. However, when using our gating module to integrate different LoRA branches, the LoRA branches cannot be merged into the pre-trained matrix at inference time, which introduces additional computational costs. Nevertheless, we demonstrate that these computational costs are minimal compared to the computational cost of the original language models (LMs).

Table 8 compares the floating-point operations (FLOPs) and multiply-add operations (MACs) during inference for different

Table 9. Results with standard deviation on different task sequences using T5-large model.

| Method | Order 1 | | Order 2 | | Order 3 | | Order 4 | |
|---|---|---|---|---|---|---|---|---|
| | AP↑ | FT↓ | AP↑ | FT↓ | AP↑ | FT↓ | AP↑ | FT↓ |
| MIGU+FT (Du et al., 2024) | - | - | - | - | $71.30_{\pm1.85}$ | $11.39_{\pm1.92}$ | $69.05_{\pm0.71}$ | $14.06_{\pm0.86}$ |
| SeqLoRA | $7.30_{\pm1.12}$ | $47.60_{\pm0.94}$ | $7.03_{\pm0.49}$ | $47.97_{\pm0.07}$ | $49.46_{\pm2.42}$ | $27.60_{\pm4.09}$ | $33.81_{\pm0.01}$ | $45.53_{\pm1.60}$ |
| IncLoRA (Hu et al., 2022) | $12.33_{\pm0.56}$ | $41.93_{\pm0.17}$ | $16.65_{\pm0.91}$ | $36.56_{\pm1.30}$ | $61.19_{\pm0.85}$ | $13.63_{\pm1.27}$ | $62.46_{\pm0.34}$ | $15.92_{\pm0.46}$ |
| C-LoRA (Smith et al., 2024) | $22.69_{\pm0.01}$ | $24.25_{\pm0.90}$ | $32.81_{\pm0.64}$ | $11.60_{\pm0.23}$ | $66.83_{\pm0.56}$ | $8.64_{\pm0.32}$ | $61.86_{\pm1.77}$ | $14.18_{\pm1.50}$ |
| O-LoRA (Wang et al., 2023a) | $26.37_{\pm2.27}$ | $19.15_{\pm2.15}$ | $32.83_{\pm0.25}$ | $11.99_{\pm0.38}$ | $70.98_{\pm1.74}$ | $3.69_{\pm0.53}$ | $71.21_{\pm0.33}$ | $4.03_{\pm1.00}$ |
| GainLoRA (O-LoRA) | $\mathbf{47.84}_{\pm0.16}$ | $\mathbf{2.26}_{\pm0.06}$ | $46.84_{\pm0.11}$ | $2.91_{\pm0.13}$ | $73.37_{\pm0.01}$ | $3.02_{\pm0.81}$ | $76.01_{\pm0.49}$ | $\mathbf{2.49}_{\pm0.12}$ |
| InfLoRA (Liang & Li, 2024) | $39.78_{\pm0.57}$ | $7.64_{\pm0.54}$ | $39.57_{\pm0.94}$ | $8.93_{\pm0.37}$ | $75.15_{\pm0.06}$ | $4.19_{\pm0.13}$ | $75.79_{\pm0.56}$ | $3.47_{\pm0.45}$ |
| GainLoRA (InfLoRA) | $46.21_{\pm0.05}$ | $2.40_{\pm0.24}$ | $46.44_{\pm0.41}$ | $\mathbf{2.61}_{\pm0.25}$ | $\mathbf{78.01}_{\pm0.26}$ | $\mathbf{0.77}_{\pm0.01}$ | $77.54_{\pm0.23}$ | $1.25_{\pm0.10}$ |

Table 10. The overall results on different task sequences with T5-XL model.

| Method | Order 1 | | Order 2 | | Order 3 | | Order 4 | |
|---|---|---|---|---|---|---|---|---|
| | AP↑ | FT↓ | AP↑ | FT↓ | AP↑ | FT↓ | AP↑ | FT↓ |
| O-LoRA (Wang et al., 2023a) | $36.50_{\pm4.29}$ | $11.42_{\pm5.30}$ | $40.64_{\pm1.09}$ | $6.37_{\pm0.66}$ | $73.77_{\pm1.14}$ | $2.70_{\pm0.54}$ | $76.19_{\pm0.49}$ | $3.56_{\pm0.40}$ |
| GainLoRA (O-LoRA) | $\mathbf{50.10}_{\pm0.22}$ | $3.21_{\pm0.32}$ | $49.86_{\pm0.06}$ | $3.04_{\pm0.13}$ | $78.41_{\pm0.50}$ | $2.59_{\pm0.56}$ | $77.21_{\pm0.19}$ | $3.30_{\pm0.34}$ |
| InfLoRA (Liang & Li, 2024) | $45.61_{\pm1.28}$ | $5.60_{\pm1.35}$ | $45.85_{\pm0.10}$ | $5.10_{\pm0.32}$ | $80.22_{\pm0.04}$ | $2.09_{\pm0.11}$ | $79.43_{\pm0.03}$ | $1.71_{\pm0.09}$ |
| GainLoRA (InfLoRA) | $50.06_{\pm0.11}$ | $\mathbf{1.86}_{\pm0.28}$ | $50.26_{\pm0.14}$ | $2.64_{\pm0.41}$ | $\mathbf{81.22}_{\pm0.11}$ | $\mathbf{0.58}_{\pm0.01}$ | $80.30_{\pm0.11}$ | $\mathbf{0.75}_{\pm0.15}$ |

models with and without GainLoRA. The computation of FLOPs and MACs follows the existing project calflops (Ye, 2023). Here, "Original" denotes the original LMs without any LoRA adaptation. Methods such as O-LoRA and InfLoRA avoid additional computational costs by merging their LoRA branches into the original weights during inference, resulting in FLOPs and MACs identical to the original LMs. Despite introducing additional FLOPs and MACs compared to the original LMs, GainLoRA maintains minimal computational overhead relative to the original LMs.

## C.2. Results with standard deviation

Table 9, Table 10 and Table 11 report the results with standard deviation.

Table 11. The overall results on different task sequences with Llama-2-7B and Llama-2-13B.

| Method | Llama-2-7B | | | | Llama-2-13B | | | |
|---|---|---|---|---|---|---|---|---|
| | Order 1 | | Order 2 | | Order 1 | | Order 2 | |
| | AP↑ | FT↓ | AP↑ | FT↓ | AP↑ | FT↓ | AP↑ | FT↓ |
| O-LoRA (Wang et al., 2023a) | $39.37_{\pm0.24}$ | $15.84_{\pm0.58}$ | $37.55_{\pm0.70}$ | $20.23_{\pm0.20}$ | $43.92_{\pm0.42}$ | $14.15_{\pm0.35}$ | $40.05_{\pm0.46}$ | $19.53_{\pm0.50}$ |
| GainLoRA (O-LoRA) | $51.10_{\pm0.91}$ | $4.96_{\pm0.56}$ | $\mathbf{51.14}_{\pm1.01}$ | $5.57_{\pm0.65}$ | $52.47_{\pm0.24}$ | $4.78_{\pm0.27}$ | $51.68_{\pm0.63}$ | $5.86_{\pm0.44}$ |
| InfLoRA (Liang & Li, 2024) | $42.93_{\pm0.77}$ | $11.23_{\pm0.24}$ | $39.94_{\pm0.30}$ | $15.00_{\pm0.51}$ | $43.64_{\pm0.02}$ | $14.85_{\pm0.31}$ | $45.74_{\pm0.81}$ | $10.61_{\pm0.09}$ |
| GainLoRA (InfLoRA) | $\mathbf{51.27}_{\pm0.01}$ | $\mathbf{2.84}_{\pm0.11}$ | $50.17_{\pm0.32}$ | $\mathbf{4.71}_{\pm0.22}$ | $\mathbf{53.64}_{\pm0.81}$ | $\mathbf{2.87}_{\pm0.05}$ | $\mathbf{52.46}_{\pm0.50}$ | $\mathbf{4.90}_{\pm0.30}$ |

## C.3. Varying the Architecture of Gating Module

### C.3.1. Varying Function $f(\cdot)$ in Gating Module

To implement our method, we define function $f(\cdot)$ as (7). Here, we vary the formula of function $f(\cdot)$ as the following two functions:

$$\min\{|b|, 1\}, \ |\sin(\frac{\pi b}{2})|. \tag{21}$$

Clearly, these two functions map real values among $[0, 1]$ and satisfy $f(0) = 0$. Table 12 shows the results. As we can see, when changing the formula of $f(\cdot)$, GainLoRA also improves the performance of O-LoRA and InfLoRA.

### C.3.2. Varying the Shapes of Weights in Gating Module

In this section, we vary the shapes of the weights in the gating modules with T5-Large. Specifically, we set the weights $\boldsymbol{W}_{i,1} \in \mathbb{R}^{d_h \times 1024}$ and $\boldsymbol{W}_{i,2} \in \mathbb{R}^{1024 \times d_h}$ in each gating module $g_i(\cdot)$ and vary $d_h$ over $\{50, 100, 200\}$. Figure 6 (a) and Figure 6 (b) show the results. As we can see, when increasing $d_h$, the performance of GainLoRA remains relatively stable,

*Table 12.* Varying the function $f(\cdot)$ in GainLoRA on different task sequences with T5-large model.

| Method | Order 1 | | Order 2 | |
|--------|---------|---------|---------|---------|
| | AP↑ | FT↓ | AP↑ | FT↓ |
| GainLoRA (InfLoRA) $(f(b) = \|2\text{sigmoid}(b) - 1\|)$ | $46.21_{\pm0.05}$ | $2.40_{\pm0.24}$ | $\mathbf{46.44_{\pm0.41}}$ | $2.61_{\pm0.25}$ |
| GainLoRA (InfLoRA) $(f(b) = \min\{\|b\|, 1\})$ | $45.05_{\pm0.32}$ | $2.07_{\pm0.24}$ | $45.00_{\pm0.20}$ | $\mathbf{1.74_{\pm0.44}}$ |
| GainLoRA (InfLoRA) $(f(b) = \|\sin(\frac{\pi b}{2})\|)$ | $\mathbf{47.48_{\pm0.03}}$ | $\mathbf{1.21_{\pm0.52}}$ | $45.03_{\pm0.67}$ | $2.37_{\pm0.34}$ |
| InfLoRA | $39.78_{\pm0.57}$ | $7.64_{\pm0.54}$ | $39.57_{\pm0.94}$ | $8.93_{\pm0.37}$ |
| GainLoRA (O-LoRA) $(f(b) = \|2\text{sigmoid}(b) - 1\|)$ | $47.84_{\pm0.16}$ | $\mathbf{2.26_{\pm0.06}}$ | $46.84_{\pm0.11}$ | $\mathbf{2.91_{\pm0.13}}$ |
| GainLoRA (O-LoRA) $(f(b) = \min\{\|b\|, 1\})$ | $\mathbf{49.62_{\pm0.57}}$ | $2.83_{\pm0.73}$ | $\mathbf{48.62_{\pm0.47}}$ | $3.74_{\pm0.02}$ |
| GainLoRA (O-LoRA) $(f(b) = \|\sin(\frac{\pi b}{2})\|)$ | $48.49_{\pm0.92}$ | $3.84_{\pm0.54}$ | $47.20_{\pm0.85}$ | $4.69_{\pm0.65}$ |
| O-LoRA | $26.37_{\pm2.27}$ | $19.15_{\pm2.15}$ | $32.83_{\pm0.25}$ | $11.99_{\pm0.38}$ |

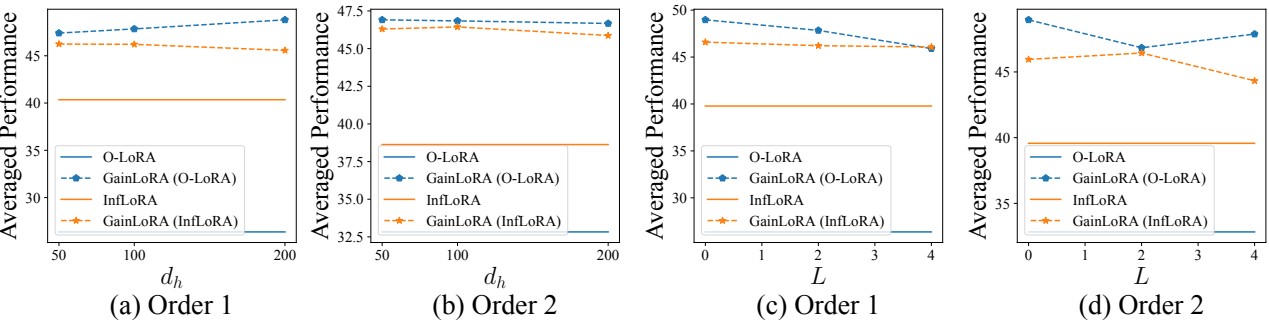

(a) Order 1      (b) Order 2      (c) Order 1      (d) Order 2

*Figure 6.* (a) and (b) show the variation of our methods' performance with the shapes of the weights in the gating module. (c) and (d) show the variation of our methods' performance with the Layers of the gating module.

indicating that our method is robust to the shape of the weights in the gating module. Note that the number of trainable parameters increases as $d_h$ increases.

### C.3.3. VARYING THE LAYERS OF GATING MODULE

In this section, we vary the layers of the gating modules with T5-Large. Specifically, we vary across $\{0, 2, 4\}$. when $L = 0$, there is only one layer with $\boldsymbol{W}_{i,1} \in \mathcal{R}^{1\times1024}$ in each gating module $g_i(\cdot)$. When $L = 2$, there are three layers with $\boldsymbol{W}_{i,1} \in \mathcal{R}^{100\times1024}$, $\boldsymbol{W}_{i,1} \in \mathcal{R}^{1024\times100}$ and $\boldsymbol{W}_{i,1} \in \mathcal{R}^{1\times1024}$. When $L = 4$, there are 5 layers with $\boldsymbol{W}_{i,1} \in \mathcal{R}^{100\times1024}$, $\boldsymbol{W}_{i,1} \in \mathcal{R}^{1024\times100}$, $\boldsymbol{W}_{i,1} \in \mathcal{R}^{100\times1024}$, $\boldsymbol{W}_{i,1} \in \mathcal{R}^{1024\times100}$, and $\boldsymbol{W}_{i,1} \in \mathcal{R}^{1\times1024}$ in each gating module. Figure 6 (c) and Figure 6 (d) show the results. As we can see, when increasing the layers of gating modules, the performance of GainLoRA remains relatively stable, indicating that our method is robust to the layers of the gating module. Note that the number of trainable parameters increases as the number of layers in gating modules increases.

### C.4. Varying Ranks in LoRA Branches

In this section, we vary the rank of LoRA branches across $\{2, 4, 8\}$ with T5-Large. Figure 7 shows the results. As shown, when the rank of LoRA branches increases, the performance of GainLoRA remains relatively stable. Note that the number of trainable parameters increases as the rank of LoRA branches increases.

### C.5. Adopting Other Update Strategies for the New LoRA Branch

Our GainLoRA does not impose specific update strategies for the new LoRA branches. In this work, we adopt the same update strategies as the existing two methods, O-LoRA (Wang et al., 2023a) and InfLoRA (Liang & Li, 2024). Related methods, such as IncLoRA (Hu et al., 2022) and C-LoRA (Smith et al., 2024), also adopt the expandable LoRA architecture illustrated in Figure 1 and fix all integration coefficients $\{a_i\}_{i=1}^{T}$ to 1. Our method GainLoRA can also adopt their update strategies for the new LoRA branch. Table 13 presents the results, demonstrating that GainLoRA further improves the

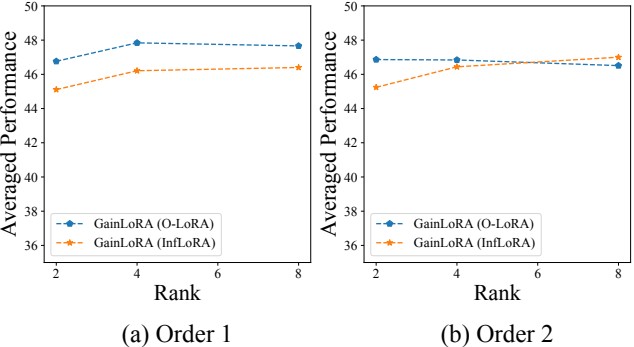

(a) Order 1            (b) Order 2

*Figure 7.* The variation of our methods' performance with the Layers of the gating module.

*Table 13.* The overall results on different task sequences with T5-large model.

| Method | Order 1 | | Order 2 | |
|--------|---------|----|---------|----|
| | AP↑ | FT↓ | AP↑ | FT↓ |
| IncLoRA | $12.33_{\pm 0.56}$ | $41.93_{\pm 0.17}$ | $16.65_{\pm 0.91}$ | $36.56_{\pm 1.30}$ |
| GainLoRA (IncLoRA) | $47.82_{\pm 0.08}$ | $3.73_{\pm 0.25}$ | $\mathbf{45.42}_{\pm 1.19}$ | $\mathbf{5.83}_{\pm 1.53}$ |
| C-LoRA | $22.69_{\pm 0.01}$ | $24.25_{\pm 0.90}$ | $32.81_{\pm 0.64}$ | $11.60_{\pm 0.23}$ |
| GainLoRA (C-LoRA) | $\mathbf{49.24}_{\pm 0.21}$ | $\mathbf{2.94}_{\pm 0.41}$ | $46.23_{\pm 0.61}$ | $6.05_{\pm 0.51}$ |

performance of these two methods.

## C.6. Extending to the Rehearsal Setting

In this work, we focus on the non-rehearsal setting, where no real or synthetic samples from old tasks are available during the learning of a new task. In this section, we demonstrate that our method, GainLoRA, can also be extended to the rehearsal setting. Specifically, in the rehearsal setting, a set of samples $\mathcal{N}_t$ containing real or synthetic samples from the previous $t-1$ tasks is available while the model learns the $t$-th new task. In this case, the constraints introduced in Section 3.2.1 are no longer necessary, and we can optimize the following loss function:

$$\frac{1}{|\mathcal{D}_t|} \sum_{(\boldsymbol{x}_t, \boldsymbol{y}_t) \in \mathcal{D}_t} \sum_{j=1}^{|\boldsymbol{y}_t|} \log\left[P(y_{t,j}|\boldsymbol{x}_t, y_{t,1}, ..., y_{t,j-1})\right] + \sum_{(\boldsymbol{x}, \boldsymbol{y}) \sim \mathcal{N}_t} \log g_t(\boldsymbol{x}), \tag{22}$$

The second term in (22) minimizes the contribution from the new LoRA branch on old tasks.

We compare GainLoRA with SAPT-LoRA (Zhao et al., 2024). For a fair comparison, we use the same rehearsal dataset as SAPT-LoRA, generated using a trained generative model. As shown in Table 14, GainLoRA achieves comparable performance to SAPT-LoRA in the rehearsal setting. Note that SAPT-LoRA is specifically designed for the rehearsal setting and is not applicable to the non-rehearsal setting, which is considered in this work.

*Table 14.* The overall results on Order 1 in the rehearsal-setting.

| | T5-Large | | Llama-2-7B | |
|--------|----------|-----|------------|-----|
| | AP↑ | FT↓ | AP↑ | FT↓ |
| SAPT-LoRA (Zhao et al., 2024) | $51.38_{\pm 0.12}$ | $0.74_{\pm 0.18}$ | $55.88_{\pm 0.25}$ | $0.74_{\pm 0.27}$ |
| GainLoRA (InfLoRA) + Replay | $51.62_{\pm 0.56}$ | $0.08_{\pm 0.10}$ | $55.93_{\pm 0.69}$ | $0.95_{\pm 0.39}$ |

## C.7. Scaling to Unseen Tasks

We further follow existing work (Zhao et al., 2024) and select 3 tasks from each task category in SuperNI benchmark to assess the model's cross-task generalization ability. The selected datasets are shown in Table 15. Table 16 shows the results.

*Table 15.* Details of selected unseen tasks in SuperNI Benchmark.

| Dataset name | Task Type | Metric |
|---|---|---|
| Task360_spolin_yesand_response_generation | summarization | Rouge-L |
| Task574_air_dialogue_sentence_generation | summarization | Rouge-L |
| Task1714_convai3_sentence_generation | summarization | Rouge-L |
| Task180_intervention_extraction | information extraction | Rouge-L |
| Task678_ollie_actual_relationship_answer_generation | information extraction | Rouge-L |
| Task1410_dart_relationship_extraction | information extraction | Rouge-L |
| Task339_record_answer_generation | dialogue generation | Rouge-L |
| Task670_ambigqa_question_generation | dialogue generation | Rouge-L |
| Task1327_qa_zre_answer_generation_from_question | dialogue generation | Rouge-L |
| Task522_news_editorial_summary | question answering | Rouge-L |
| Task1356_xlsum_title_generation | question answering | Rouge-L |
| Task1499_dstc3_summarization | question answering | Rouge-L |
| Task421_persent_sentence_sentiment_classification | sentiment analysis | Accuracy |
| Task833_poem_sentiment_classification | sentiment analysis | Accuracy |
| Task929_products_reviews_classification | sentiment analysis | Accuracy |

Here, 'Sum', 'IE', 'Dialogue', 'QA' and 'SA' denote the summarization tasks, information extraction tasks, dialogue tasks, question answering tasks and sentiment analysis tasks, respectively. Our methods yield better overall performance than other methods.

*Table 16.* The results of different methods on unseen tasks after training on Order 1 with Llama-2-7B model.

| Method | Sum | IE | Dialogue | QA | SA | Avg |
|---|---|---|---|---|---|---|
| O-LoRA | 6.77 | 36.53 | 31.79 | 14.66 | 61.67 | 30.28 |
| GainLoRA (O-LoRA) | 6.03 | **42.97** | **44.52** | 15.34 | **75.44** | **36.86** |
| InfLoRA | 8.72 | 31.33 | 36.51 | **19.63** | 61.55 | 31.55 |
| GainLoRA (InfLoRA) | **8.94** | 34.35 | 38.74 | 19.19 | 64.22 | 33.09 |

