# OpenReview forum: "Gated Integration of Low-Rank Adaptation for Continual Learning of Language Models"
_ICML.cc/2025/Conference — Submitted to ICML 2025_

### Official Review · Reviewer_QgmP · 2025-03-03

**Overall Recommendation:** 3

**Summary:**

This manuscript focused on the continual learning of language models. Unlike the existing continual learning studies based on LoRA that treated the new and old LoRA branches to contribute equally to old tasks, the authors proposed a new method, gated integration of low-rank adaptation (GainLoRA). Specifically, GainLoRA expands a new LoRA branch for a new task with a gating module to integrate. The introduced gating modules are used to integrate the new and old branches, and the new gating module will minimize the contribution from the new LoRA branch to old tasks to mitigate the forgetting issue. Experiments were conducted on several language benchmarks with two language models to support the effectiveness of the proposed method.

### update after rebuttal and internal discussion

Thanks to the authors for responding to my questions. After the rebuttal, most of my concerns have been addressed. However, during the internal discussion, there are some discussions regarding the completeness of the comparisons with many existing studies in the field of continual learning with pre-trained model (including but not limitted to S-prompt/HiDe-prompt/RanPAC/Dual-prompt/NoGRA/HiDe-PET). Based on this consideration, I will keep my original rating "weak accept".

**Claims And Evidence:**

Yes, the claims were supported by either theoretical analysis and empirical results.

**Essential References Not Discussed:**

N/A

**Experimental Designs Or Analyses:**

The experimental designs and analyses have been checked and they seem sound.

**Methods And Evaluation Criteria:**

The benchmarks used in this manuscript are reasonable for evaluation.

**Other Comments Or Suggestions:**

The authors should also consider the introduced memory and computation by the subspace construction.

**Other Strengths And Weaknesses:**

## Strengths
1. The design of gating modules is reasonable and theoretically sound.
2. The proposed GainLoRA can be plugged into other existing LoRA-based methods to provide further improvements.
3. The experiments were conducted on the real-world datasets and models, like T5 and Llama2, which were at the scale in production environments.
## Weaknesses
1. For the computational and memory overhead, the authors only emphasize the associated amount introduced by trainable parameters. However, the memory and computational overhead regarding the subspace construction should also be discussed.

**Questions For Authors:**

1. In the current version, it seems that the authors mainly discussed the computational and memory overhead by trainable parameters. However, to the best of my experience in using Gradient Projection Memory (GPM) and its variants, I noticed that the construction of subspaces $\mathcal{M}$ and the projection operations can also require significant computation and memory. I wonder if the authors could provide quantitative discussions regarding this aspect.
2. In Eq. (12), it seems that $l \in \{1,2,...,T-1\}$ should be  $l \in \{i,i+1,...,T-1\}$.
3. Does the gating module introduce significant training instability, particularly in early task learning phases?
4. Is your proposed method compatible with SAPT [1], a recent CL method for LLMs with parameter-efficient tuning? If so, could you please provide performance comparisons in the experimental part?

I will accordingly adjust the rating after the author rebuttal.

References:

[1] SAPT: A Shared Attention Framework for Parameter-Efficient Continual Learning of Large Language Models. ACL 2024.

**Relation To Broader Scientific Literature:**

The core contribution in this manuscript is the design of gating modules. This design can be inspired by the related topics like Mixture-of-Expert. However, it is still novel to see the use of gating modules in the LoRA-based continual learning problem.

**Theoretical Claims:**

The demonstration of the conditions of the gating module seems correct.

---

> ### Author Rebuttal · Authors · 2025-04-01
>
> **Q1: memory and computational overhead regarding the subspace construction**
>
> **A1:** The memory and computational overhead of subspace construction in GainLoRA is minimal due to the small size of the gating module (only 3 layers, see Appendix B.3). We provide detailed analyses below.
>
> Memory: The number of orthogonal bases stored for each subspace does not exceed its dimension. For T5-Large, the dimensions of the three subspaces are 1024, 100, and 1024, respectively. This results in a worst-case memory of less than 0.3% of the total model parameters ($(2*1024^2+100^2)$/(T5-Large's params)<0.3%). Similar estimates yield 0.07%, 0.5%, and 0.4% for T5-XL, Llama-2-7B, and Llama-2-13B, respectively. Since this calculation represents a rough upper bound, the actual memory is even lower.
>
> Computational Overhead: The computational overhead for subspace construction requires a single forward pass over the task dataset and SVD on the feature matrices of the gating module.
>
> Assuming a single forward pass over the task dataset requires $A$ FLOPs. For T5-Large, training a task for 100 epochs needs 100 forward and backward passes. Since a single backward pass has roughly $2A$ FLOPs, the total FLOPs are $300A$. Thus, a single forward pass for subspace construction accounts for only 1/300≈0.33% of total computation. Similar estimates yield 0.33%, 0.67%, and 0.67% for T5-XL, Llama-2-7B, and Llama-2-13B, respectively.
>
> SVD is performed on $H_lH_l^T\in\mathbb{R}^{d_l\times d_l}$, where $H_l$ is the feature matrix in the $l$-th layer of the gating module. According to the conclusion from Lecture 31 of the textbook [1], the FLOPs required for the SVD of $H_lH_l^T$ are less than $4d_l^3$. For T5-Large ($d_1=d_3=1024$ and $d_2=100$), this results in $4*(2*1024^3+100^3)<5GFLOPs$, which is negligible compared to a single forward pass with sequence length 128 (see Table 8 in the Appendix). Similar calculations give the same conclusion for T5-XL, Llama-2-7B, and Llama-2-13B.
>
> We will include these calculations in the final version. Thanks for suggestions.
>
> **Q2: projection operations can also require significant computation**
>
> **A2:** The projection operations in Eq.9 and Eq.10 incur minimal computation. Before learning a new task, Eq.9 is applied to the last layer of the gating module, involving at most three matrix multiplications. During training, after a single forward-backward pass, Eq.10 is applied to all layers of the gating module, involving at most nine matrix multiplications. In contrast, a single forward-backward pass of T5 or LLaMA involves hundreds or even thousands of matrix multiplications. Therefore, the computation of projection operations is negligible compared to the overall training process. We will include these analyses in the final version. Thanks for suggestions.
>
> **Q3: $l\in 1,2,...,T-1$ should be $l\in i,i+1,...,T-1$.**
>
> **A3:** We follow existing works [2,3] to define FT and have verified that the correct formulation is indeed $l \in 1,2,...,T-1$ as stated in their papers.
>
> **Q4: Does the gating module introduce significant training instability, particularly in early task learning phases?**
>
> **A4:** No, the gating module does not introduce significant training instability. This [figure](https://anonymous.4open.science/r/Re-A3CF/track.png) shows the variation in the gating module's output during training on the 15-th new task in Order 1. As observed, the output for new tasks quickly approaches 1, ensuring sufficient adaptation to new tasks without unstable training. Meanwhile, the output for old tasks remains near 0, maintaining stability for old tasks.
>
> **Q5: Is your proposed method compatible with SAPT ...?**
>
> **A5:** Our method is partially compatible with SAPT but cannot be directly integrated with it. SAPT relies on generated samples for rehearsal, making it incompatible with the rehearsal-free setting considered in this work. Furthermore, SAPT requires an extra phase to train a generative model for producing old samples, involving multiple forward and backward passes over the task dataset. This leads to significantly higher computational overhead compared to our GPM-based subspace construction, a key concern raised by the reviewer.
>
> When rehearsal is allowed, GainLoRA doesn't need to constrain the new gating module but uses generated old samples to minimize its output on old tasks. The following table shows the results of Order 1, where GainLoRA uses the same rehearsal datasets as SAPT and achieves comparable performance. However, SAPT can't be extended to a rehearsal-free setting, while GainLoRA is designed for rehearsal-free setting. We've cited SAPT and will incorporate this discussion in the final version. Thanks for the suggestions.
> ||T5-Large|Llama-2-7B
> :-|:-:|:-:
> SAPT-LoRA|51.38|55.88
> GainLoRA+rehearsal|51.62|55.93
>
> [1] Numerical linear algebra, SIAM 2022
>
> [2] Continual Learning in Low-rank Orthogonal Subspaces, NeurIPS 2020
>
> [3] On tiny episodic memories in continual learning, arXiv 2019

---

> > ### Comment · Reviewer_QgmP · 2025-04-02
> >
> > Thanks for providing further explanations to my questions. Most of my concerns have been addressed. I decided to increase my rating to Accept./

---

> > > ### Author Response · Authors · 2025-04-03
> > >
> > > We are pleased to see that your key concerns have been effectively addressed. We sincerely appreciate your time and effort in reviewing our response and providing positive feedback.

---

### Official Review · Reviewer_kvg3 · 2025-03-12

**Overall Recommendation:** 3

**Summary:**

This paper introduces GainLoRA, which integrates LoRA with gating mechanisms. GainLoRA expands a new LoRA branch for each task while incorporating task-specific gating modules, for mitigating catastrophic forgetting. Experimental results demonstrate strong performance and provide comprehensive ablations.

**Claims And Evidence:**

Yes

**Essential References Not Discussed:**

Wu, et al. Mixture of lora experts. ICLR2024.

**Experimental Designs Or Analyses:**

yes

**Methods And Evaluation Criteria:**

yes

**Other Comments Or Suggestions:**

This work is complete and sound, though the novelty feels limited.

**Other Strengths And Weaknesses:**

Strengths:
1. The paper is straightforward and easy to follow.
2. The experiments and ablations are comprehensive, addressing many key concerns.
3. The improvement over prior works is significant, demonstrating better performance in mitigating forgetting.

Weaknesses:
1. The idea of using a mixture of LoRA branches is not novel, as it closely resembles the MoE LoRA framework. The primary contribution appears to be its application to the continual learning domain, with added constraints on gate learning.
2. The paper uses $W$ to represent both the pre-trained weight matrices added with LoRA and the weights of the gating module, which could be misleading.
3. The proposed soft-gating mechanism (sigmoid-based gating) diverges from the commonly used top-k gating, raising concerns about scalability. For instance, in the SuperNI benchmark with 1616 tasks, even using a low-rank of 4 would lead to an explosion in the number of parameters during both training and inference. This issue becomes more critical when considering computational throughput and latency constraints.
4. The gradient projection memory method may struggle with extremely long task sequences, as orthogonal subspaces are inherently limited. Over time, maintaining orthogonality across a growing number of tasks could degrade performance.
5. The initialization strategy in Eq. 8, which copies the gating weights from the previous task, may introduce a conflict with the desired orthogonal property.

**Questions For Authors:**

See weakness.

**Relation To Broader Scientific Literature:**

The method aligns with recent trends in parameter-efficient fine-tuning in continual learning.

**Theoretical Claims:**

yes

---

> ### Author Rebuttal · Authors · 2025-04-01
>
> **Q1: The idea of using a mixture of LoRA branches is not novel, as it closely resembles the MoE LoRA framework.**
>
> **A1:** Our method is fundamentally different from existing MoE LoRA frameworks, as it specifically addresses continual learning (CL) in a rehearsal-free setting where task identities are unavailable during inference. While MoE-based LoRA frameworks dynamically route inputs across experts, they do not tackle catastrophic forgetting or adapt to sequentially arriving tasks without rehearsal. In contrast, our method leverages gating mechanisms to minimize the interference of the new LoRA branch on old tasks.
>
> Furthermore, the paper mentioned by the reviewer, Wu et al., Mixture of LoRA Experts (ICLR 2024), does not focus on CL or address forgetting. Instead, it targets at static learning with MoE-style routing, which is fundamentally different from our setting. We will cite this paper and include a discussion in the final version, clarifying the distinctions between our approach and existing MoE-based LoRA methods. Thanks for suggestions.
>
> **Q2: The paper uses $W$ to represent both the pre-trained weight matrices added with LoRA and the weights of the gating module, which could be misleading.**
>
> **A2:** In the final version, we will use $G_{l}$ to represent the weight of the $l$-th layer in the gating module, ensuring a clear distinction from the pre-trained weight matrices. Thanks for suggestions.
>
> **Q3: The proposed soft-gating mechanism diverges from the commonly used top-k gating, raising concerns about scalability. For instance, in the SuperNI benchmark with 1616 tasks, ... lead to an explosion in the number of parameters during both training and inference...**
>
> **A3:** To the best of our knowledge, the number of tasks in the 15-task sequence setting in our experiments matches or exceeds the scale of nearly all existing CL methods for language models (LMs). Our results demonstrate that under this setting, our method introduces minimal additional parameters and computational overhead. Notably, since no existing CL method for LMs has explored a sequence with more than 15 tasks, making a direct jump from 15 to 1616 seems too challenging for the development of the CL community.
>
> We acknowledge that as the number of tasks increases, the parameter count and computational cost of our method will also grow. However, in an extremely long task sequence, constraining CL methods to avoid parameter growth can lead to insufficient capacity for learning new tasks, which could ultimately degrade performance. Therefore, scaling CL methods to extremely long task sequences remains an open problem. As part of future work, we plan to investigate top-k gating or other adaptive mechanisms to enhance efficiency while preserving performance. We will highlight this issue in the final version. Thanks for suggestions.
>
> **Q4: The gradient projection memory method may struggle with extremely long task sequences, as orthogonal subspaces are inherently limited. Over time, maintaining orthogonality across a growing number of tasks could degrade performance.**
>
> **A4:** Performance degradation over long task sequences is a well-known challenge in continual learning (CL) and is not unique to our method. To overcome forgetting, many methods introduce constraints like regularization or orthogonal constraints. As old tasks accumulate, these constraints must be strengthened, which can limit model plasticity and degrade new task performance. Reducing constraints can mitigate performance degradation on new tasks but may risk forgetting old tasks. This trade-off, known as the plasticity-stability dilemma, is inherent to CL and affects all methods, including those using orthogonal subspaces.
>
> In fact, although our method introduces orthogonal subspaces, it mitigates performance degradation because the orthogonal subspaces are applied to the gating module rather than directly to the LoRA parameters. This allows the model to avoid excessive constraints on the LoRA parameters, thereby preserving its ability to learn new tasks.
>
> We will provide a more detailed discussion in the final version. Thanks for suggestions.
>
> **Q5: The initialization strategy in Eq. 8 may introduce a conflict with the desired orthogonal property.**
>
> **A5:** The initialization strategy in Eq. 8 does not conflict with the desired orthogonal property. This is because it only copies the gating weights of the first L layers from the previous task and does not copy their updates. Since Eq. 6 applies to weight updates rather than the weights themselves, there is no conflict between the initialization strategy in Eq. 8 and the desired orthogonal property in Eq. 6.
>
> Furthermore, while Eq. 5 is applied to the gating weights in the last layer, Eq. 8 does not initialize this layer’s weights from the previous task. Therefore, there is no conflict between the initialization strategy in Eq. 8 and the desired orthogonal property in Eq. 5.

---

### Official Review · Reviewer_UCPw · 2025-03-12

**Overall Recommendation:** 3

**Summary:**

The paper introduces GainLoRA, an approach to mitigate catastrophic forgetting in task incremental continual learning scenarios leveraging gated integration of low-rank adapters. This approach expands LoRA branches for each task and introduces gating modules to dynamically control the impact of each branch. Unlike the previous approaches that integrate LoRA branches with naive averaging, GainLoRA computes integration coefficients regarding each LoRA’s contributions to the input data to better adapt to where task identities are unavailable.

**Claims And Evidence:**

**Well-supported** \
*Claim 1: GainLoRA improves CL (continual learning) performance by mitigating catastrophic forgetting and outperforms SOTA methods*
- The authors experimentally shows that the proposed GainLoRA approach achieves lower forgetting rates (FT) and higher averaged performance (AP) compared to existing SOTA methods, such as O-LoRA and InfLoRA

*Claim 2: The gating module ensures dynamic task adaptation*
- The outputs of gating module illustrated in Figure 5 demonstrates that the gating module assigns higher coefficients to the newly added task
- Table 4 ablation study showcases that the task orthogonal property of gating modules mitigate the forgetting issue

**Partially supported** \
*Claim 3: GainLoRA has minimal computational overhead*
- GainLoRA stores a separate LoRA branch and gating module per task, requiring dynamic computation of integration coefficients for each input sample.
- While the LoRA branches are lightweight, the cumulative parameter count increases with the number of tasks, potentially limiting scalability in real-world applications where models must handle a large number of tasks across diverse domains.

**Essential References Not Discussed:**

The paper provides a strong foundation by referencing LoRA-based CL methods such as O-LoRA and InfLoRA, but it omits several recent rehearsal-free CL methods such as MoCL, TaSL, KIF that are highly relevant for comparison. See "Methods And Evaluation Criteria".

**Experimental Designs Or Analyses:**

**Strengths**
- The selection of benchmarks (SuperNI, Long Sequence) and comparison methods (O-LoRA, C-LoRA, InfLoRA, etc.) is well-grounded. The experiments are appropriately designed for task-incremental CL scenarios, and the ablation studies are thorough and well-executed.

**Weaknesses**
- Analysis of task order impacts should be included, particularly to assess how the orthogonality constraint in the gating module affects knowledge transfer. Evaluating potential negative transfer risks when tasks are highly similar would strengthen the paper’s insights.
- Including multi-task learning (MTL) results would provide an upper bound on performance, offering a benchmark to assess how well GainLoRA retains task performance and mitigates forgetting compared to joint training on all tasks.

**Methods And Evaluation Criteria:**

**Strengths**
- GainLoRA is evaluated on standard CL datasets, including SuperNI and Long Sequence.
- The AP and FT metrics are well-established for measuring continual learning effectiveness.
- GainLoRA's robustness is tested across multiple task orders.

**Weaknesses**
- Additional CL evaluation metrics (FWT, BWT) should be included to provide a more comprehensive assessment of GainLoRA’s ability to enhance knowledge transfer and reduce forgetting.
- Task order diversity is not fully explored beyond random sequences. Specifically, evaluating highly similar consecutive tasks (e.g., sentiment analysis → sentiment analysis) is crucial, as the orthogonality constraint may inadvertently hinder knowledge transfer in such cases.
- Comparisons with recent SOTA rehearsal-free baselines for language models, such as [MoCL (2024)](https://aclanthology.org/2024.naacl-short.39/), [TaSL (2024)](https://aclanthology.org/2024.acl-long.69/), and widely adopted continual learning baselines like [EWC (2017)](https://www.pnas.org/doi/10.1073/pnas.1611835114) would provide a more comprehensive evaluation of GainLoRA’s effectiveness.

**Other Comments Or Suggestions:**

I find the paper well-written, methodologically sound, and a valuable extension of LoRA-based continual learning, with the gating mechanism for task-adaptive LoRA integration being a particularly noteworthy contribution. However, the paper would benefit from a broader analysis of task order effects on gating modules, additional CL evaluation metrics (FWT, BWT), and comparisons with recent SOTA CL methods beyond LoRA-based approaches. Additionally, including MTL results would provide an upper bound on performance, offering a stronger reference for evaluating GainLoRA’s effectiveness. If these concerns are addressed, I would be willing to raise my score.

**Other Strengths And Weaknesses:**

**Strengths**
- The paper is clearly written with a well-structured presentation, making the proposed GainLoRA method easy to understand and follow.
- GainLoRA is straightforward to implement and can be seamlessly integrated with existing LoRA-based continual learning approaches.
- Evaluation across multiple model scales (T5, Llama-2-7B, Llama-2-13B) and diverse task orders ensures robust empirical validation.

**Weaknesses**
- While its application to LoRA integration is innovative, the gating module itself is not inherently novel, as similar gating mechanisms have been explored in PEFT and CL methods.
- Potential negative transfer effects from the gating module and its orthogonality constraint are not analyzed—specifically, how these constraints impact task similarity, transfer learning, and overall adaptation remains unexplored.

**Questions For Authors:**

1. How does GainLoRA mitigate negative transfer when continually learning highly similar tasks? Would removing or relaxing the orthogonality constraint improve performance for similar tasks?
2. How does the output distribution of the gating module vary based on task order? Does the size of each task dataset influence the numerical values of the output coefficients (e.g., lower gating weights for low-resource tasks)?

**Relation To Broader Scientific Literature:**

- Catastrophic forgetting is a critical issue in continual learning, particularly for domain adaptation and test-time adaptation in real-world applications.
- GainLoRA contributes to LoRA-based parameter-efficient continual learning, extending prior methods such as O-LoRA and InfLoRA.
- Its effectiveness in mitigating forgetting without relying on task identities enhances its practical applicability, making it more adaptable for real-world scenarios where task boundaries are ambiguous or unknown.

**Theoretical Claims:**

The mathematical formulation of gating function and orthogonality constraints are well-defined and theoretically grounded.

---

> ### Author Rebuttal · Authors · 2025-04-01
>
> **Q1: the cumulative parameter count increases ... potentially limiting scalability in ... a large number of tasks**
>
> **A1:** We admit that cumulative parameters increase with more tasks, but scaling to a large number of tasks remains a challenge in CL. To the best of our knowledge, our 15-task sequence setting matches or exceeds those in nearly all existing CL methods for LMs, including TaSL, EWC, and MoCL. Experiments show our method adds minimal additional parameters and computational overhead in this setting.
>
> From a capacity perspective, our method scales better than fixed-capacity methods like O-LoRA and InfLoRA since they may lead to insufficient capacity over a large number of tasks. In contrast, our method slightly expands capacity per task. We'll incorporate these discussions into the final version. Thanks for suggestions.
>
> **Q2: FWT, BWT should be included**
>
> **A2:** This [table](https://anonymous.4open.science/r/Re-A3CF/T1.png) reports FWT and BWT for different methods. Our method achieves the best BWT by preventing new LoRA branches from interfering with old tasks.
>
> For FWT, our method is competitive. This is because old gating modules generate coefficients for new task samples on old branches. Since we do not enforce 0 outputs from old gating modules for new task samples, new samples can leverage old LoRA branches' knowledge. Note that we do not claim FWT improvement, and our focus is on enhancing overall CL performance by reducing forgetting, as shown by our best FT and BWT. We'll include these results and discussions in the final version. Thanks for suggestions.
>
> **Q3: Comparisons with MoCL, TaSL, EWC**
>
> **A3:** For TaSL and EWC, this [table](https://anonymous.4open.science/r/Re-A3CF/T3.png) shows that our methods outperform them in AP and FT. This [table](https://anonymous.4open.science/r/Re-A3CF/T1.png) also shows their FWT and BWT.
>
> For MoCL, we maintain the same settings as those in MoCL, including 16-shot, 4 tasks, and 3 different orders. After adjusting the update magnitude (see **A7**), this [table](https://anonymous.4open.science/r/Re-A3CF/T4.png) shows that our methods outperform MoCL in the setting where task identities are unavailable during testing.
>
> We will cite these references in the final version and make discussions. Thanks for suggestions.
>
> **Q4: Including MTL results**
>
> **A4:** In the response to Reviewer kvg3 (**A2**), we provide MTL results. These will be added to the final version. Thanks for suggestions.
>
> **Q5: the gating module itself is not inherently novel**
>
> **A5:** We admit that the gating module has been used before, but in rehearsal-free setting where task identities are unavailable during testing, we are the first to explore how to design gating modules to overcome forgetting. Note that this setting is important and has been considered by many CL methods such as OLoRA and TaSL.
>
> **Q6: Evaluating potential negative transfer risks when tasks are highly similar... (e.g., sentiment analysis→sentiment analysis)**
>
> **A6:** We evaluated GainLoRA on similar consecutive tasks (3 sentiment analysis tasks: Task363→Task1687→Task875) as suggested. The results in the [table](https://anonymous.4open.science/r/Re-A3CF/T2.png) show that while GainLoRA remains effective, its improvement is smaller than that in the 15-task setting with dissimilar tasks. Furthermore, GainLoRA underperforms InfLoRA and O-LoRA on the new task (Task875) but outperforms them on old tasks (Task363 and Task1687). This indicates orthogonality constraints might hinder forward transfer. This is a common trade-off in CL: constraints help mitigate forgetting but may restrict transfer, particularly for similar tasks. Conversely, weak or no constraints risk forgetting in dissimilar tasks. Future work will explore adaptive strategies: stronger constraints for dissimilar tasks and weaker ones for similar tasks. Anyway, our GainLoRA is effective in terms of overall (average) accuracy. These discussions and results will be added in the final version. Thanks for this insightful comment.
>
> **Q7: How does the output distribution of the gating module vary based on task order? lower gating weights for low-resource tasks?**
>
> **A7:** The output distribution of the new gating module is not significantly affected by task order, as shown in Figure 5 of the text and in this [figure](https://anonymous.4open.science/r/Re-A3CF/Sim.png), which involves a task order with similar tasks mentioned in **A6**.
>
> Low-resource tasks may result in lower gating weights, but adjusting the gating module's update magnitude can mitigate this. Specifically, for 16-shot experiments (see **A3**), as shown in the [figure](https://anonymous.4open.science/r/Re-A3CF/Few.png), using small learning rate as in many-shot settings leads to insufficient learning, resulting in small coefficients for new tasks and poor performance. However, increasing the learning rate boosts both the new task coefficients and model performance.

---

> > ### Comment · Reviewer_UCPw · 2025-04-04
> >
> > Thank you for the detailed and thoughtful responses. The additional experiments and analyses, including broader baselines and FWT/BWT metrics, have addressed my main concerns. While some limitations remain (e.g., novelty, scalability), the paper is acceptable as it offers a promising approach for rehearsal-free continual learning scenarios. I will adjust my rating to Weak Accept.

---

> > > ### Author Response · Authors · 2025-04-04
> > >
> > > Thank you for your thoughtful review and constructive feedback. We respectfully clarify that we offers a novel method specifically for the rehearsal-free continual learning setting—an important setting in continual learning.
> > >
> > > Regarding scalability, our method is more flexible in terms of model capacity: unlike fixed-capacity methods (e.g., O-LoRA, InfLoRA) that may suffer from capacity insufficiency as tasks accumulate, our design incrementally expands capacity with minimal per-task parameter growth. This enables better adaptation to a large number of tasks.
> > >
> > > We are pleased that our responses have addressed your main concerns, and we sincerely appreciate your time and effort in reviewing our work and providing positive feedback.

---

### Official Review · Reviewer_C736 · 2025-03-13

**Overall Recommendation:** 4

**Summary:**

The paper proposes a method for computing the weighting factor of different LoRA components in a continual learning setting. The approach is based on training a new set of LoRA parameters for each new task alongside a gating network. This network is constructed such that it outputs a value of 0 at 0. The method further enforces orthogonality constraints to prior data at initialization and when updating to avoid interference with old tasks. The experiments show improved performance over baselines from the literature on SuperNI and LongSequence benchmarks.

**Claims And Evidence:**

The primary contribution of the paper is a new method and it performs best in the experiments.

**Essential References Not Discussed:**

None that I am aware of.

**Experimental Designs Or Analyses:**

The design of the experiments is suitable. There are further relevant ablation studies on the individual components of the method.

**Methods And Evaluation Criteria:**

Yes, the benchmarks, metrics and baselines are appropriate.

**Other Comments Or Suggestions:**

* the constraint to not carry any data forward seems a bit artificial to me when a new lora module + gate function is added for each task (hence memory use scale linearly with the number of tasks anyway).
* please include results for simultaneously training on all tasks as a reference for optimal performance where appropriate, e.g. in Tab 1
* I would be curious if there are any variants of the method that didn't work? If so it would be helpful to include these in the appendix.

**Other Strengths And Weaknesses:**

The paper leverages prior work in a thoughtful and logical way. The method is evaluated thoroughly both in comparison to prior work and in terms of ablations. I could see future work extend on this paper.

**Questions For Authors:**

n/a

**Relation To Broader Scientific Literature:**

The paper clearly credits the works on orthogonal initialization and updating that it builds on. Wider themes in the related literature (parameter efficient fine-tuning, continual learning) are similarly discussed and reference.

**Theoretical Claims:**

No.

---

> ### Author Rebuttal · Authors · 2025-04-01
>
> **Q1: the constraint to not carry any data forward seems a bit artificial to me when a new lora module + gate function is added for each task (hence memory use scale linearly with the number of tasks anyway).**
>
> **A1** The constraint to not carry any data forward is not merely about saving memory but also about preserving privacy and reducing computational overhead [1]. In many real-world scenarios, storing past data is not allowed due to privacy concerns, making rehearsal-free methods essential. Furthermore, some methods that generate pseudo-data for replay require training a generative model, which incurs significant computational overhead. On the contrary, rehearsal-free methods like our GainLoRA in this paper inherently avoid these issues. In the final version, we will clarify this in more detail. Thanks for the suggestion.
>
> **Q2: please include results for simultaneously training on all tasks as a reference for optimal performance where appropriate, e.g. in Tab 1**
>
> **A2:** We conduct simultaneously training on all tasks in SuperNI and Long Sequence, and we refer to this method as multi-task learning (MTL). The average performance is reported in the table below, and we will include these results in Table 1, Table 2 and Table 3 in the final version. Thanks for the suggestion.
>
> ||SuperNI|Long Sequence
> |:-|:-:|:-:|
> |T5-Large|52.10|81.63
> |T5-XL|54.12|84.07
>
> ||SuperNI
> |:-|:-:|
> |Llama-2-7B|56.88
> |Llama-2-13B|57.66
>
> **Q3: I would be curious if there are any variants of the method that didn't work? If so it would be helpful to include these in the appendix.**
>
> **A3:** Yes, we explored several variants of our method that did not perform well, and we reported them in the ablation study (Table 4). Specifically, the variant "No Initialization Constraints" replaces $f$ with a sigmoid function, a common choice for gating mechanisms. However, sigmoid function does not satisfy $f(0)=0$, leading to performance degradation compared to our method. Similarly, the variant "No Update Constraints" omits orthogonal projection during training, which also results in a significant performance drop. We appreciate the reviewer’s suggestion and will clarify these points further in the final version.
>
> [1] A comprehensive survey of continual learning: Theory, method and application, TPAMI 2024.

---

> > ### Comment · Reviewer_C736 · 2025-04-08
> >
> > Thank you for the clarifications and additional results. These further assure me in my recommendation to accept the paper.

---

> > > ### Author Response · Authors · 2025-04-09
> > >
> > > We are pleased that our responses have addressed your concerns, and we sincerely appreciate your time and effort in reviewing our work and providing positive feedback.

---

### Decision · Program_Chairs · 2025-05-01

**Decision:**

Reject

**Comment:**

The paper proposes an interesting approach for continual learning with pre-trained models, with applications to NLP tasks. However, I find the paper has not positioned nor compared with a rather substantial & highly visible literature on continual learning with pre-trained models. To give a few examples:

S-prompt: https://arxiv.org/abs/2207.12819

HiDe-prompt: https://arxiv.org/abs/2310.07234

RanPAC: https://arxiv.org/abs/2307.02251

Dual-prompt: https://arxiv.org/abs/2204.04799

NoGRA: https://arxiv.org/abs/2405.14124

HiDe-PET (generalization of HiDe-prompt): https://arxiv.org/abs/2407.05229

Some of these are generalized variants of the L2P method that the paper compared with. While these methods were tested on computer vision datasets, the proposed techniques are mostly task- and model-agnostic so not discussing and not comparing with them unfortunately make the claimed contribution inconclusive.

For example, why can't we repurpose existing continual learning techniques with pre-trained model to solve the NLP tasks in this paper? If we can, are they effective? To motivate the development of new techniques, I feel that it is important to establish empirically or at least provide detailed discussion that reusing existing techniques is not possible in the targeted scenarios, which motivates the development of new techniques.

--

I also note that the statement in lines 154-163 is not fully supported by the experiment as was claimed. For example, CoDA-prompt (Smith et al., 2023b) was cited but not compared with.

--

Overall, I feel that the literature review of this work is missing a substantial body of relevant work here. While I am aware that this concern was unfortunately not raised during the author-reviewer discussion, I feel that the required amount of empirical experiments that will be needed to properly address this concern is beyond the scope of the rebuttal due to its missing of a large no. of previous work.